# Agent S: An Open Agentic Framework that Uses Computers Like a Human

**Saaket Agashe**,* **Jiuzhou Han**,* **Shuyu Gan, Jiachen Yang, Ang Li, Xin Eric Wang**
Simular Research

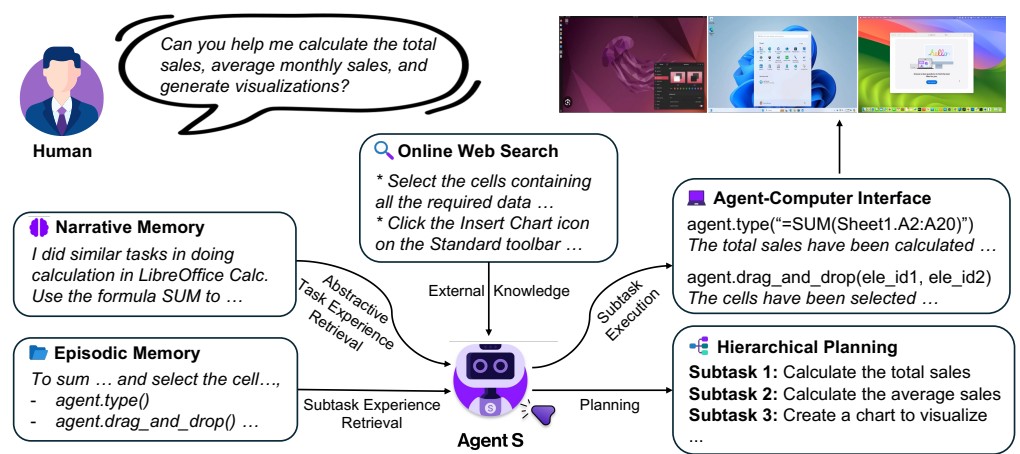

Figure 1: Agent S uses a computer like a human to solve diverse desktop tasks on different systems.

## Abstract

We present Agent S, an open agentic framework that enables autonomous interaction with computers through a Graphical User Interface (GUI), aimed at transforming human-computer interaction by automating complex, multi-step tasks. Agent S aims to address three key challenges in automating computer tasks: acquiring domain-specific knowledge, planning over long task horizons, and handling dynamic, non-uniform interfaces. To this end, Agent S introduces experience-augmented hierarchical planning, which learns from external knowledge search and internal experience retrieval at multiple levels, facilitating efficient task planning and subtask execution. In addition, it employs an Agent-Computer Interface (ACI) to better elicit the reasoning and control capabilities of GUI agents based on Multimodal Large Language Models (MLLMs). Evaluation on the OSWorld benchmark shows that Agent S outperforms the baseline by 9.37% on success rate (an 83.6% relative improvement) and achieves a new state-of-the-art. Comprehensive analysis highlights the effectiveness of individual components and provides insights for future improvements. Furthermore, Agent S demonstrates broad generalizability to different operating systems on a newly-released WindowsAgentArena benchmark. Code available at https://github.com/simular-ai/Agent-S.

## 1 Introduction

*"The digital revolution is far more significant than the invention of writing or even of printing."*

*— Douglas Engelbart, The Inventor of Computer Mouse*

Since its invention, the mouse has been controlled by humans for interacting with computers. But does it really have to be? Autonomous Graphical User Interface (GUI) agents offer the promise of

---

*Equal contributions.

solving very specific and highly varied user queries—such as data entry, scheduling, and document creation for individual users, and streamlining operations in commercial settings—in the most general way: through direct UI interaction using the mouse and keyboard. Moreover, by eliminating the need for constant manual interaction, these agents not only boost efficiency but also improve accessibility, empowering individuals with disabilities to interact with technology in new, transformative ways. Recent advancements in Multimodal Large Language Models (MLLMs), such as GPT-4o (OpenAI, 2023) and Claude (Anthropic, 2024), have laid the foundation for the development of GUI agents for human-centred interactive systems like desktop OS (Xie et al., 2024; Bonatti et al., 2024).

However, automating computer tasks presents significant challenges. First, the vast range of constantly-evolving applications and websites requires the agent to possess specialized and up-to-date domain knowledge and the ability to learn from open-world experience. Second, complex desktop tasks often involve long-horizon, multi-step planning with interdependent actions that must be executed in a specific sequence. The agent must, therefore, create a clear plan with intermediate subgoals and track task progress. Third, GUI agents must navigate dynamic, non-uniform interfaces, processing large volumes of visual and textual information while operating within a vast action space. This involves distinguishing between relevant and irrelevant elements, accurately interpreting graphical cues, and responding to visual feedback during task execution.

In this paper, we present **Agent S**, a new agentic framework that tackles these challenges towards the goal of using computers like a human. First, to enhance the GUI agent's capabilities in solving diverse, long-horizon desktop tasks with specific domain knowledge, we propose an *Experience-Augmented Hierarchical Planning* method. This approach leverages Online Web Knowledge and past experiences stored in Narrative Memory to decompose the complex, long-horizon task into a structured plan of manageable subtasks (see Figure 1). Online Web Knowledge provides up-to-date external knowledge about specific applications, allowing the agent to adapt to frequently changing software and websites. Narrative Memory contains high-level, abstractive task experiences from past interactions, equipping the agent with contextual understanding for effective task planning. The agent monitors task completion progress, and during each subtask execution, it retrieves detailed, step-by-step subtask experience from Episodic Memory to dynamically refine its actions and continuously improve its planning ability. Successful subtasks and the full task experience are evaluated, summarized, and stored in episodic and narrative memory to enable continual improvement.

Furthermore, we introduce a specific language-centric *Agent-Computer Interface (ACI)* (Lieberman & Selker, 2003) as an abstraction layer to improve grounding, safety, and efficiency for MLLM-based GUI agents. The ACI defines an interaction paradigm by (1) *a dual-input strategy* using visual input for understanding environmental changes together with an image-augmented accessibility tree for precise element grounding; (2) *a bounded action space* of language-based primitives (e.g., `click(element_id)`) that are conducive to MLLM common-sense reasoning and generate environment transitions at the right temporal resolution for the agent to observe immediate and task-relevant environment feedback.

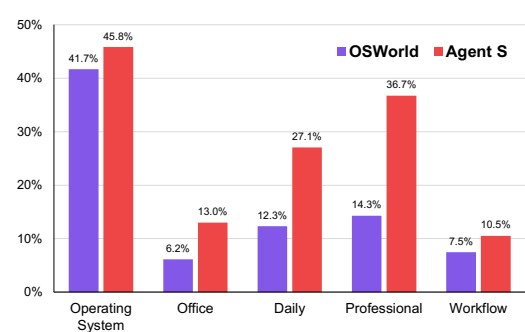

Figure 2: Agent S *vs.* OSWorld Agent results across five broad computer task categories.

Our approach shows a remarkable improvement in the overall performance of Agent S on the OSWorld benchmark (OpenAI, 2023) (from 11.21% to 20.58%, with a relative improvement of 83.6%), establishing the new state-of-the-art results. The detailed comparison is shown in Figure 2, which demonstrates consistent improvements by Agent S across five broad computer task categories over the OSWorld agent. We also evaluate our Agent S on a concurrent work—WindowsAgentArena (Bonatti et al., 2024), where we observe a performance improvement from 13.3% to 18.2% on an equivalent setup without any explicit adaptation. The improvement demonstrates the broad generalizability of Agent S to different operating systems. We detail the component-wise improvements introduced by the proposed strategies through ablation studies and

present a comprehensive error analysis of our Agent S framework. In summary, our contributions are four-fold:

- We introduce Agent S, a new agentic framework that integrates experience-augmented hierarchical planning, self-supervised continual memory update, and an Agent-Computer Interface for MLLM-based GUI agents to perform complex computer tasks.
- We propose an experience-augmented hierarchical planning method that uses experience from external web knowledge and the agent's internal memory to decompose complex tasks into executable subtasks.
- We extend the concept of an ACI to GUI agents, allowing MLLM-based agents to operate computers more precisely using a set of high-level, predefined primitive actions.
- We conduct experiments on different operating systems in the OSWorld and WindowsAgentArena benchmarks, where the results show that Agent S achieves new state-of-the-art performance.

## 2   RELATED WORK

**MLLM Agents.** The advent of Multimodal Large Language Models (MLLMs) has led to a host of works that utilize them as a reasoning backbone in Agentic Systems (Sumers et al., 2024). These Agents augment LLMs with Memory, Structured Planning (Wang et al., 2023; Shinn et al., 2023; Weng et al., 2023), Tool Use (Schick et al., 2023; Shen et al., 2023; Patil et al., 2023) and the ability to Act in external environments Park et al. (2023). These agents have shown promise in domains ranging from embodied simulators (Liang et al., 2023; Song et al., 2023) to video games (Wu et al., 2023; Wang et al., 2024) and scientific research (Bran et al., 2023). For Software Engineering (Hong et al., 2024; Qian et al., 2024) in particular, Yang et al. (2024) proposed an Agent-Computer Interface (Lieberman & Selker, 2003) for MLLM agents to understand and act more efficiently and reliably. Our work extends and integrates these individual modules into a new MLLM agent framework for computer control.

**GUI Agents.** MLLM agents have been applied to execute natural language instructions in both web and OS environments. Early research concentrated on web navigation tasks, utilizing MLLMs to interact with web interfaces (Gur et al., 2024; He et al., 2024; Kim et al., 2023; Shaw et al., 2023; Putta et al., 2024). Recently, the focus has shifted to OS-level environments, leading to the development of benchmarks and frameworks such as OSWorld Xie et al. (2024) and WindowsAgentArena Bonatti et al. (2024) for desktop control, and DiGIRL (Bai et al., 2024) and AndroidWorld (Rawles et al., 2024) for mobile environments. These OS-level tasks offer broader control capabilities beyond the limitations of single-browser contexts in web navigation. Methodologically, earlier GUI agents employed behavioral cloning with reinforcement learning (Humphreys et al., 2022), in-context trajectory examples (Zheng et al., 2024b), state-dependent offline experience (Fu et al., 2024b), and reusable skill generation (Wang et al., 2024). Contemporaneous work on GUI agents for video games and OS (Wu et al., 2024; Song et al., 2024; Tan et al., 2024) propose varying instances of cognitive architectures (Sumers et al., 2024). Our work contributes unique modules such as experience-augmented hierarchical planning and ACI for GUI control, integrated with a novel continual memory update framework.

**Retrieval-Augmented Generation (RAG) for AI Agents.** RAG (Fan et al., 2024) improves the reliability of MLLM inference by augmenting the input with reliable and up-to-date external knowledge. Similarly, MLLM agents benefit from retrieving task exemplars (Kim et al., 2024), state-aware guidelines (Fu et al., 2024a), and past experiences (Kagaya et al., 2024). Our use of experience for augmentation differs in three ways: 1) our hierarchical planning leverages both full task experience and subtask experience; 2) the full task experience is summarized into an abstractive textual reward for subtask planning; 3) the subtask experience is assessed and annotated by a self-evaluator before being stored in memory.

## 3   AGENT S

The GUI-based Operating System control tasks can be formalized as a **Partially Observable Markov Decision Process (POMDP)**, defined as $\mathcal{M} = (\mathcal{S}, \mathcal{O}, \mathcal{A}, \mathcal{T}, \mathcal{R})$, where $\mathcal{S}$ is the OS state space, $\mathcal{O}$ is the observation space (natural language instructions, screenshots, accessibility trees,

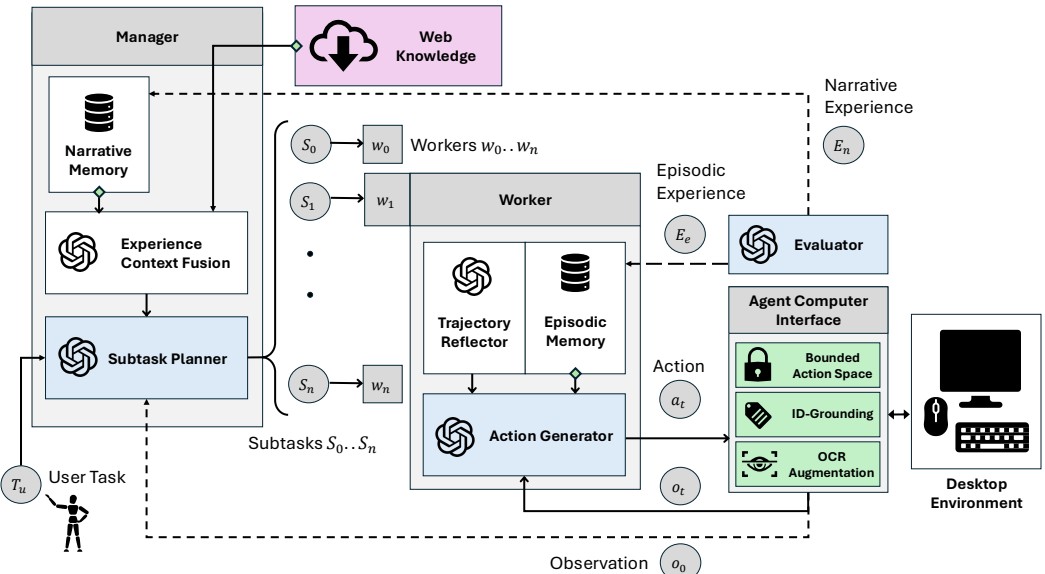

Figure 3: Overview of the Agent S framework. Given task $T_u$ and initial environment observation $o_0$, the Manager conducts experience-augmented hierarchical planning using web knowledge and narrative memory to produce subtasks $s_0, \ldots, s_n$. For each $s_i$, Worker $w_i$ draws from episodic memory to generate an action $a_t$ at time $t$, which is executed by the ACI to return the next immediate observation $o_{t+1}$. A self-evaluation module closes the loop by storing the summarized subtask and full-task trajectories in narrative and episodic memory.

etc.), $\mathcal{A}$ is the action space (clicks, keys, etc), $\mathcal{T} : \mathcal{S} \times \mathcal{A} \rightarrow \mathcal{S}$ is the state transition function, and $\mathcal{R} : \mathcal{S} \times \mathcal{A} \rightarrow [0, 1]$ is the reward function. Agent S, illustrated in Figure 3, is a novel framework that integrates three main strategies in a closed loop to tackle such complex GUI-based operating system control tasks: experience-augmented hierarchical planning, continual update of narrative and episodic memory, and an Agent-Computer Interface for precise perception and action on GUIs. Experience-augmented hierarchical planning allows Agent S to break down complex tasks into manageable subtasks. This enables both high-level planning and low-level execution to draw from external web-based experience and internal task-specific experience. A continual process of storing and retrieving self-evaluated task experience in narrative and episodic memory enables Agent S to improve over time and adapt to changes in the open-world desktop environment. The ACI ensures grounding by providing a vision-augmented accessibility tree observation containing all valid GUI elements and constraining the agent's chosen action to a bounded discrete space of valid actions. Below, we describe each component and its integration in detail.

## 3.1 Experience-augmented Hierarchical Planning

In hierarchical planning *Manager* decomposes a high-level task $T$ into a sequence of subtasks $\{S_1, S_2, \ldots, S_n\}$, where each subtask $S_i$ is more granular and feasible for execution. The Manager assigns these subtasks to *Workers*, which execute them by performing low-level actions. Each Worker operates in a localized decision-making loop, selecting actions at each timestep based on its observation. The hierarchical structure allows the Manager to focus on planning while Workers handle execution.

### 3.1.1 Manager: Fusing External Knowledge and Internal Experience for Planning

The Manager $G$ is the primary plan generator module in our system. It receives a task $T_u$ from the user and the initial environment observation $O_0$ (Annotated Accessibility Tree + Screenshot) from the ACI as input. The manager formulates an observation-aware query $Q$ based on the user instruction and its observation in a "How to do X" format. This query is used for two types of

retrieval. First, the query is used for **Online Web Search** through Perplexica Search Engine[1] to get external knowledge. Then the same query is used to retrieve a similar task experience summary from the Manager's own **Narrative Memory** $M_n$. The retrieval is based on the similarity of the query embedding.

The Narrative Memory includes summaries of both successful and failed trajectories with specific actions removed as *abstractive full task experience* $E_{n_u}$. The success/failure is evaluated by the Self-Evaluator $S$ module (described in Subsection 3.1.3) without any human feedback or ground truth information. This two-step retrieval provides the Manager with both the general and specific domain knowledge required to plan for the task. The outputs of the retrieval process are fused into a single fused guideline using the **Experience Context Fusion** submodule, represented formally as:

$$Q = \text{LLM}(T_u, O_0), \qquad K_{\text{web}} = \text{Retrieve}(\text{Web}, Q)$$
$$E_{n_u} = \text{Retrieve}(M_n, Q), \quad K_{\text{fused}} = \text{LLM}(M_n(Q), K_{\text{web}})$$

The fused knowledge $K_{\text{fused}}$ is then used by **Subtask Planner** submodule of the Manager to formulate a detailed, topologically sorted queue of subtasks $\langle s_0...s_n \rangle$ that can accomplish the user instruction. The manager also generates associated context $C_{s_i}$ for each subtask $s_i$ which includes additional information useful to accomplish the subtask.

### 3.1.2 WORKER: LEARNING FROM SUBTASK EXPERIENCE AND TRAJECTORY REFLECTION

The subtasks $\langle s_0..s_n \rangle$ generated by the Manager $G$ are executed sequentially by Worker modules $\langle w_0..w_n \rangle$. Each Worker can take multiple time steps within one episode to complete a subtask $s_i$. Firstly, the combination of the User Task $T_u$, the subtask $s_i$ and the contextual information $C_{s_i}$ are used as a query to retrieve similar subtask experience $E_{s_i}$ from the Worker's **Episodic Memory**. The Episodic Memory is indexed by the concatenation of the task query, the subtask, and the contextual information $\langle Q, s_i, C_{s_i} \rangle$, based on the similarity of the embedding. As opposed to Narrative Memory, Episodic Memory includes a complete plan with specific grounding actions and only summaries from the subtask trajectories designated as DONE or successful by a Worker. Additionally, a **Trajectory Reflector** submodule $TR_i$ is associated with each worker. This submodule observes the entire episode as the worker is executing the subtask and provides reflective advice to the agent—helping it think of alternative strategies and avoid repetitive actions.

$$E_{s_i} = \text{Retrieve}(M_e, \langle T_u, s_i, C_{s_i} \rangle)$$

The subtask experience $E_{s_i}$ and the reflection is used by the **Action Generator** submodule inside a Worker to generate a single structured response - consisting of a previous action status check, observation analysis, semantic next action and grounded next action. This structured response allows the agent to generate a templated chain-of-thought Wei et al. (2022); Yao et al. (2023) for improved reasoning and results in a single grounded action $a_j$. This action is passed to the ACI which implements it in the Desktop Environment. Once the worker reasons that the subtask has been completed, it generates a special grounded action DONE which signals the successful end of the subtask. The worker can also optionally generate a FAIL signal, in which case the hierarchical operation is reset and the Manager replans a new set of subtasks based on the intermediate environment configuration.

### 3.1.3 SELF-EVALUATOR: SUMMARIZING EXPERIENCES AS TEXTUAL REWARDS

The Self-Evaluator $S$ is responsible for generating experience summaries as textual rewards $r$ for the Manager and Worker modules. In the case of the successful end of an episode signaled by the Worker with a DONE signal, the evaluator observes the complete episode and generates learning in the form of a summarization of the strategy used by the worker to complete that subtask. This strategy is fed back into the Worker's episodic memory $M_e$. In the case of the end of the complete user-provided task, indicated either by the successful completion of all subtasks or by the maximum number of steps limit, the evaluator generates a learning signal in the form of the summary of the entire task completion process. This summary is fed back and saved in the narrative memory $M_n$ of the Manager. This process of Observations, Hierarchical Action Generation, and Rewards in the form of textual summaries to update the internal memories of the Manager and Worker mirrors a classic Hierarchical Reinforcement Learning process - but uses Retrieval as a learning strategy.

---

[1] https://github.com/ItzCrazyKns/Perplexica

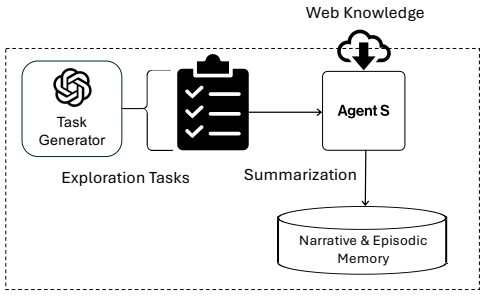

(a) Self-supervised Exploration

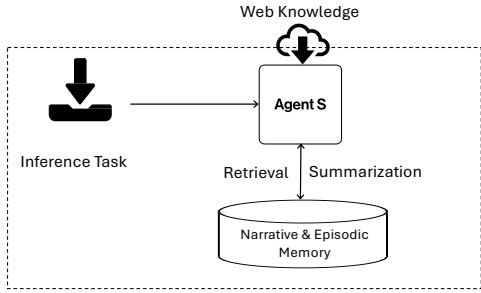

(b) Continual Memory Update

Figure 4: The pipeline of memory construction and update, which contains two phases: Self-supervised Exploration and Continual Memory Update. The initial Narrative & Episodic Memory is constructed through some randomly curated tasks during the exploration phase, and then it is updated based on the inference tasks continually.

## 3.2 MEMORY CONSTRUCTION AND UPDATE

**Initial Memory Construction via Self-supervised Exploration.** To bootstrap Narrative $M_n$ and Episodic Memories $M_e$, Agent S conducts self-supervised exploration on a set of synthetically generated tasks (see Figure 8). We utilize two methods to create two types of random exploration tasks: environment-independent tasks and environment-aware tasks. For environment-independent tasks, we leverage a task generator to generate the top 50 most common tasks from the various applications used in OSWorld (Xie et al., 2024) and WindowsAgentArena (Bonatti et al., 2024). For environment-aware tasks, we take the initial environments of the tasks in OSWorld and WindowsAgentArena and prompt a Task Generator to generate a different task based on the environment. Both types of tasks consist of the exploration tasks. Then we run Agent S on these tasks by only taking web knowledge $K_{\text{web}}$ and collect the full task (Narrative Experience $E_n$) and subtask experiences (Episodic Experience $E_e$) for the narrative and episodic memories. The key stored in narrative memory $M_n$ is the query $Q$ and for episodic memory $M_e$, the key is query $Q$ concatenated with subtask information $\langle Q, s_i, C_{s_i} \rangle$. Through this process, the initial memory is constructed.

**Continual Memory Update.** As our Agent S interacts with new tasks, it continually updates the Narrative Memory $M_n$ and Episodic Memory $M_e$, as illustrated in Figure 8. Thus even after the initial exploration is completed, the agent continues to learn as it encounters and attempts newer, more novel tasks. This process enables our agent to learn even during inference and retrieve the learned knowledge to new tasks effectively.

## 3.3 AGENT-COMPUTER INTERFACE

Current desktop environments are designed to accommodate two distinct user types: (1) *human users*, who can perceive and respond to subtle visual changes in real-time, and (2) *software programs*, which execute predefined tasks through scripts and Application Programming Interfaces (APIs). However, these interfaces are inadequate for MLLM agents tasked with GUI control and manipulation at the fundamental keyboard-mouse level. These agents operate on a different paradigm: they respond in slow, discrete time intervals, lack an internal coordinate system, and cannot efficiently process fine-grained feedback after each minor mouse movement or keyboard input. Drawing inspiration from the ACI developed for Software Engineering agents (Yang et al., 2024), we propose the creation of a novel ACI to bridge the gap between the unique operational constraints of MLLM agents and the requirements of open-ended GUI-control tasks.

**Perception and Grounding.** Grounding in GUIs is the task of identifying the target element $e$, based on the natural language description of the element $\tilde{e}$. Agents need to understand this natural language directive and find the exact UI element corresponding to this description. Current MLLMs can effectively reason about certain elements and features in an image, but they cannot directly ground and pinpoint specific elements in images as they lack an internal coordinate system. In GUI manipulation, agents need to constantly interact with fine UI elements, and previous works have shown that grounding is a significant bottleneck in these agents (Xie et al., 2024; Zheng et al., 2024a). Desktop environments, however, provide an easily parseable Accessibility Tree with

coordinate information about almost every element in the UI. Thus, our ACI design incorporates a dual-input strategy with different purposes for each input. The image input is used by the agent to observe salient details about the environment—such as popups, button states, checking if a previous action worked, and reasoning about the next step. The accessibility tree input is used for reasoning about the next step and, more importantly, grounding specific elements in the environment. To achieve this, we tag each element in the accessibility tree with unique integer tags which can be used by agents when referring to these elements. Furthermore, while previous works seek to augment the image with information from the accessibility tree (Xie et al., 2024; Zheng et al., 2024a; Bonatti et al., 2024) using Set-of-Mark Prompting, we augment the tree with details from the image. To achieve this, we run an OCR module on the image and parse textual blocks from the screenshot. We then add these blocks to the accessibility tree as interactable UI elements if they do not already exist in the tree. To check for existing elements, we perform an IOU (Intersection over Union) match with all elements in the tree.

**Constrained Action Space with Concurrent Feedback.** Desktop automation has traditionally relied on APIs and scripts, but adopting these as actions would imply an unbounded combinatorial action space of arbitrary executable code. This is unsuitable for keyboard-mouse-level GUI automation agents because it compromises safety and precision. Code blocks can contain multiple sequential actions, leaving the agent with neither control over nor feedback from individual steps. To ensure that actions generated by agents are safely and reliably relayed to the environment and produce clear and timely feedback, our ACI design incorporates a bounded action space. This space includes primitive actions like click, type, and hotkey (detailed in Appendix A.1). Agents can refer to different elements by their tagged IDs, and the ACI translates the $\langle$ primitive - ID $\rangle$ information into executable Python code. Furthermore, the agent is allowed to perform only one discrete action at each time step, so it can observe immediate feedback from the environment. These actions are also coarse enough to account for the slow, stateless nature of MLLMs, e.g., the agent can directly move to and click an element instead of moving the mouse in small increments.

## 4 EXPERIMENTS

### 4.1 EXPERIMENTAL SETUP

**Benchmarks.** We evaluate Agent S on OSWorld (Xie et al., 2024), a benchmark for testing the multimodal agents' capability of executing a wide range of computer tasks in a real computer environment. This executable environment allows free-form keyboard and mouse control of real computer applications, including OS, Office (LibreOffice Calc, Impress, Writer), Daily (Chrome, VLC Player, Thunderbird), Professional (VS Code and GIMP), and Workflow (tasks involving multiple apps). In addition, we also evaluate the generalization of Agent S on WindowsAgentArena (Bonatti et al., 2024), a contemporaneous benchmark in the Windows operating system.

**Settings & Baselines.** Since the OSWorld benchmark contains 369 tasks on Ubuntu, for the backbone model of Agent S, we leverage GPT-4o and Claude-3-Sonnet, respectively. For WindowsAgentArena, we test all 154 tasks on GPT-4o. We use the PaddleOCR[2] toolkit as our OCR tool in augmenting accessibility trees for grounding. The embedding model for the retrieval we use is *text-embedding-3-small*. Agent S takes the accessibility tree and screenshot as inputs, so we also use the reported results in OSWorld (Xie et al., 2024) and WindowsAgentArena (Bonatti et al., 2024) with same input setting as baselines. The OSWorld baseline MMAgent takes the accessibility tree and screenshots as input for spatial grounding to generate the action with coordinates at each step. The WindowsAgentArena baseline NAVI (Bonatti et al., 2024) utilizes an accessibility tree, OCR, and Proprietary models to process the screenshot and create Set-of-Marks as input. Its action space includes a constrained set of primitives but allows multiple actions to be chained together.

### 4.2 MAIN RESULTS

**OSWorld.** Table 1 shows the performance comparison between Agent S and the baseline models, evaluated across the whole OSWorld test set. We compare with MMAgent Xie et al. (2024), CRADLE Tan et al. (2024), Open Interpreter, and the FRIDAY AgentWu et al. (2024). For the GPT-4o

---

[2]https://github.com/PaddlePaddle/PaddleOCR

Table 1: Main results of Successful Rate (%) on the OSWorld full test set of all 369 test examples. All agents except CRADLE use Accessibility Tree + Screenshot as inputs. FRIDAY and OpenInterpreter were evaluated by third parties.

| Agent | MLLM | OS | Office | Daily | Profess. | Workflow | Overall |
|---|---|---|---|---|---|---|---|
| MMAgent | Claude-3-opus | 12.50 | 3.57 | 5.27 | 8.16 | 1.00 | 4.41 |
| MMAgent | Gemini-Pro-1.5 | 12.50 | 3.58 | 7.83 | 8.16 | 1.52 | 5.10 |
| CRADLE | GPT-4o | 3.58 | **16.67** | 6.55 | 20.41 | 5.48 | 7.81 |
| OpenInter* | GPT-4o | - | - | - | - | - | 8.94 |
| Friday* | GPT-4o | - | - | - | - | - | 11.11 |
| MMAgent | GPT-4o | 41.67 | 6.16 | 12.33 | 14.29 | 7.46 | 11.21 |
| MMAgent | GPT-4V | 16.66 | 6.99 | 24.50 | 18.37 | 4.64 | 12.17 |
| Agent S | Claude-3.5-sonnet | 41.66 | 13.83 | **30.46** | 32.65 | 9.54 | 20.48 |
| Agent S | GPT-4o | **45.83** | 13.00 | 27.06 | **36.73** | **10.53** | **20.58** |

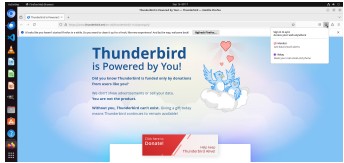 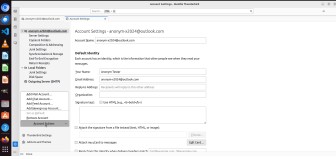 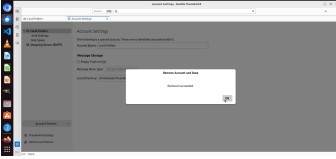

(a) Open Account Settings: *agent.click(41, 1, "left")*

(b) Remove the Account: *agent.click(86, 1, "left")*

(c) Remove the Account: *agent.click(149, 1, "left")*

Figure 5: A successful example of the Thunderbird task: "*Help me to remove the account 'anonym-x2024@outlook.com'*." For space concern, (a) (b) (c) demonstrate the screenshots, current subtasks, and grounding actions at steps 1, 4, and 6, respectively.

model, Agent S achieves an overall success rate of 20.58%, nearly doubling the performance of the best corresponding MMAgent baseline (11.21%). Agent S consistently outperforms the baselines in the "Daily" and "Professional" tasks, where it reaches 27.06% and 36.73% success rates, respectively, compared to the best baseline results of 24.50% and 18.37%. These tasks are commonly used in daily life or involved with knowledge-intensive professional applications, which benefit more from the retrieval augmentation of Agent S. Both Claude-3.5-Sonnet and GPT-4o outperform the baseline versions across the majority of tasks. The results demonstrate the enhanced capability of Agent S in handling diverse and complex tasks more effectively than the baseline approaches.

**Qualitative Examples.** In Figure 5, we illustrate an example of a task from the Thunderbird app from OSWorld: *Help me to remove the account "anonym-x2024@outlook.com"*. Agent S completes tasks by interacting with the desktop through a combination of actions. More qualitative examples are demonstrated in Appendix D.1.

## 4.3 ABLATION STUDY

To demonstrate the effectiveness of individual modules of Agent S, we stratified sampled a subset of 65 instances, $test_{sub}$[3] from the full test set for the ablation study. Considering the inference cost, we utilized GPT-4o as the LLM backbone for all ablation studies for both the baseline and Agent S.

**Learning from experience enhances the domain knowledge of GUI agents.** The Experiential learning process of Agent S involves searching web knowledge, retrieving full task experience from narrative memory and retrieving subtask experience from episodic memory. To assess the impact of different components, we systematically remove each component and observe performance changes across different task categories. The results are shown in Table 2. Learning from universal experience available as web knowledge allows Agent S to make informed plans across a wide range of tasks and has the most significant impact. The learning from Narrative and Episodic memories synergies effectively with web retrieval, and the results detail how their ablation affects the agent's ability to handle complex tasks, underscoring the value of experiential learning. These results demonstrate that each component plays a critical role in enhancing the agent's domain knowledge. Removing

---

[3]The test_small set provided by the OSWorld codebase is too small and imbalanced (only 39 examples in total and 2 in the OS category) for practical evaluations. Thus, we sample a larger and more balanced subset.

Table 2: The ablation study of experience-augmented hierarchical planning in OSWorld $test_{sub}$. The metric is Successful Rate (%).

| Method | OS (6) | Office (17) | Daily (16) | Profess. (10) | Workflow (16) | Overall (65) |
|---|---|---|---|---|---|---|
| baseline (OSWorld Agent) | 33.33 | 5.88 | 12.50 | 10.00 | 6.25 | 10.77 |
| Agent S | 50.00 | 11.76 | 37.50 | 40.00 | 12.50 | 26.15 |
| - w/o Web Knowledge | 16.60 | 11.76 | 24.49 | 30.00 | 6.25 | 16.80 |
| - w/o Narrative Memory | 33.33 | 11.76 | 36.99 | 20.00 | 12.50 | 21.41 |
| - w/o Episodic Memory | 33.33 | 5.88 | 25.00 | 30.00 | 12.50 | 18.46 |
| - w/o All | 33.33 | 5.88 | 18.75 | 20.00 | 6.25 | 13.85 |

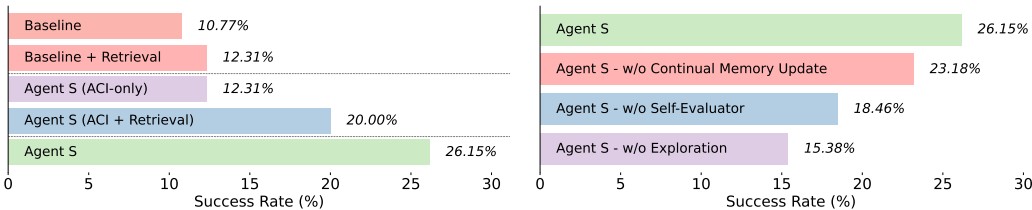

Figure 6: Ablation of ACI in OSWorld $test_{sub}$.

Figure 7: Ablation of the memory update mechanism in OSWorld $test_{sub}$.

all three components (w/o All) degrades the performance significantly, revealing the importance of *learning from experience* in the design.

**ACI elicits better reasoning abilities of LLMs and supports better agentic learning.** Figure 6 presents the results of the ablation study on the ACI module. Comparing the baseline with Agent S (ACI-only)[4] highlights the enhanced reasoning abilities achieved by incorporating ACI. Additionally, we examined the impact of ACI on agentic learning by integrating the Experiential learning process. For the baseline, adding Experiential learning slightly improved overall performance. However, when added to Agent S (ACI-only), the performance improved significantly, demonstrating ACI's effectiveness in enhancing agentic learning.

**Hierarchical Planning supports long-horizon workflows.** The (*ACI-only + Experiential Learning*) setup in Figure 6 shows Agent S performance without Hierarchical Planning, and the observed performance drop (26.15% to 20.00%) compared to the full Agent S underscores the importance of Hierarchical Planning in modeling long-horizon workflows. The effect of hierarchical formulation becomes pronounced in the presence of Experiential learning as the Manager can generate more detailed and accurate plans in the subtask planning stage.

**Exploration, Continual Memory Update and Self-Evaluator are indispensable for memory construction.** Our agent collects experience in two phases - initially during the self-supervised exploration phase and then continually as it interacts with new examples (See Figure 8). To assess the effectiveness of these two learning stages and further examine our Self-evaluator which stores experience as summaries instead of unfiltered trajectories we run the ablation shown in Figure 7. Removing exploration limits memory updates to the inference phase only. Removing the continual memory update means we only use the memory obtained from the exploration phase without subsequent updates. Removing the self-evaluator involves replacing summarized experiences with the original full trajectories. The results shown in Figure 7 reveal that ablating both the continual memory update and self-supervised exploration phases results in a performance drop, with the self-supervised exploration being much more impactful. The ablation of the Self-Evaluator further shows the benefits of using summarized trajectories instead of full trajectory exemplars for planning.

## 4.4 ERROR ANALYSIS

We performed a thorough error analysis on the tasks that Agent S failed within $test_{sub}$ of the OSWorld. There are three types of errors that we observed: (1) *Planning Error*: A planning error occurs when the agent generates unsuitable plans for a task, including inaccuracies in the plan, misleading subtask information, or misalignment of subtask sequence with task requirements. (2) *Grounding Error*: A grounding error arises when the agent fails to accurately interact with target elements

---

[4]This version of Agent S excludes Hierarchical Planning to better study the effects of ACI in isolation.

Table 3: The statistic of Error Rate (%) on $test_{sub}$ of OSWorld that Agent S failed to complete.

| Error Metric | OS | Office | Daily | Profess. | Workflow | Overall |
|---|---|---|---|---|---|---|
| Planning Error | 66.67 | 25.00 | 30.00 | 66.67 | 28.57 | 34.69 |
| Grounding Error | 0.00 | 75.00 | 50.00 | 66.67 | 35.71 | 53.06 |
| Execution Error | 33.33 | 87.50 | 100.00 | 66.67 | 71.43 | 79.59 |
| Subtask Failure | 16.67 | 58.47 | 62.82 | 33.61 | 70.43 | 57.17 |

despite their visibility and the application of correct reasoning. This includes incorrect element selection or inaccurate coordinate selection due to the inherent limitations of our action space (e.g., selecting the center instead of a more precise part of the element). (3) *Execution Error*: An execution error emerges when the agent makes incorrect decisions or fails to adjust its behavior during task execution. This includes repetitive actions, diverging from subtask goals, delays in transitioning between subtasks or violating established protocols by combining multiple actions into one.

**Statistic Results of the Errors.** We analyzed Agent S's trajectory for each failed task, identifying error types based on the definitions provided. A single task may contain multiple errors. We also calculated the Subtask Failure Rate, which measures the average percentage of failed subtasks relative to total attempts, and the Error Rate, which reflects the proportion of tasks exhibiting a specific error type. As shown in Table 3, execution and grounding errors are the most common across various task categories. A case study of error occurrence can be found in Appendix D.2.

## 4.5 GENERALIZATION TO DIFFERENT OPERATING SYSTEMS

We test the Agent S framework with no modification on WindowsAgentArena (Bonatti et al., 2024), a Windows OS benchmark released contemporaneously with our work. We compare Agent S with the similar configuration with GPT-4o as the MLLM backbone, Accessibility Tree + Image as the input, and parsing with OCR. As shown in Table 4, Agent S outperforms the Navi agent without any adaptation to the new Windows environment.

Table 4: Results of Successful Rate (%) on WindowsAgentArena using GPT-4o and Image + Accessibility Tree input on the full test set of all 154 test examples.

| Method | Office | Web Browser | Windows System | Coding | Media & Video | Windows Utils | Overall |
|---|---|---|---|---|---|---|---|
| NAVI(Bonatti et al., 2024) | 0.0 | **20.0** | 29.2 | 9.1 | **25.3** | 0.0 | 13.3 |
| Agent S | 0.0 | 13.3 | **45.8** | **29.2** | 19.1 | **22.2** | **18.2** |

## 5 CONCLUSION

In this work, we present Agent S—A novel framework for developing fully Autonomous Graphical User Interface (GUI) agents that can perform a wide range of user queries by directly controlling the keyboard and mouse. Through the Agent S framework, we show the benefits of Learning from Experience for Task-oriented GUI agents. We also discuss the concept of an Agent Computer Interface for the GUI domain, arguing in favour of an abstraction layer that allows MLLM agents to perceive and reason at a language level with rich and continuous feedback. By leveraging Experience-Augmented Hierarchical Planning, Online Web Knowledge, and an Agent-Computer Interface (ACI), Agent S demonstrates SOTA performance on the OSWorld benchmark and generalizability across different operating systems. We demonstrate the potential of MLLM agents to learn from external sources and through direct interaction with the environment, without any human or environmental feedback in the GUI agents domain, thus opening a discourse on zero-shot, agentic methods for GUI agents.

**Future Work.** A key metric that has been unaddressed in existing work on MLLM agents for computer control, including ours, is the number of agent steps and wall clock time required for task completion. While our work focuses on achieving significant improvement in task performance, future work can consider a shortest-path navigation formulation of GUI control and evaluate the Pareto-optimality of various agents on the dimensions of time and accuracy. In our work, we use the state-of-the-art GPT-4o and Claude-3.5-sonnet models. However, future work can extend the ideas of experiential learning and Agent Computer Interface for smaller, open-source MLLMs which could be fine-tuned to bridge the gap.

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

# A   AGENT-COMPUTER INTERFACE

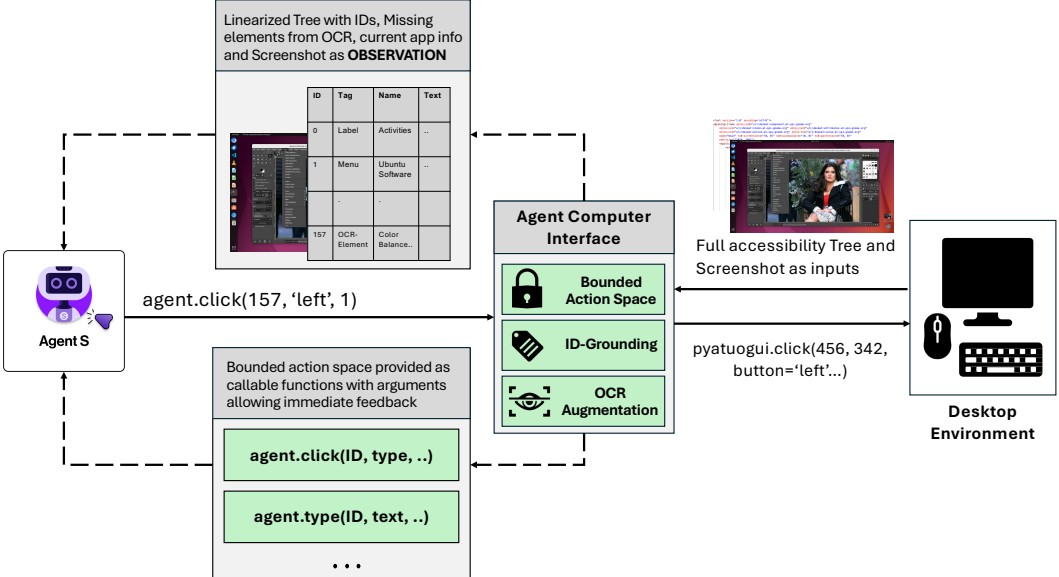

Figure 8: The Agent-Computer Interface (ACI) for Agent S integrates a linearized accessibility tree, screenshot-based observations, and a bounded action space for GUI interaction. It enables the agent to ground elements using unique integer tags and augment the accessibility tree with OCR-parsed elements from screenshots. Actions are restricted to discrete primitives (e.g., `click`, `type`) to ensure precise and safe interactions with the desktop environment, providing immediate feedback after each step.

## A.1   BOUNDED ACTION SPACE

To facilitate the agent's accurate and effective task execution, we define a constrained action space, which simplifies the action selection process, making it easier for the agent to ground its decisions in a well-structured set of operations. As summarized in Table 5, each action type has certain parameters and detailed in description.

Table 5: Agent Action Space, Descriptions, and Arguments.

| Agent Action | Action Details | |
| --- | --- | --- |
| | Description | Arguments |
| click | Click on an element. | *element_id, num_clicks, button_type, hold_keys* |
| type | Type text into an element. | *text, element_id, overwrite, enter* |
| scroll | Scroll within an element. | *element_id, clicks* |
| hotkey | Press a hotkey combo. | *keys* |
| hold_and_press | Hold keys and press others. | *hold_keys, press_keys* |
| drag_and_drop | Drag and drop between elements. | *drag_from_id, drop_on_id, hold_keys* |
| save_to_buffer | Save text to a buffer for later use. | *text* |
| switch_applications | Switch to another app. | *app_code* |
| wait | Wait for some time. | *time* |
| done | Mark task as success. | None |
| fail | Mark task as failure. | None |

## A.2   ABLATIONS ON AGENT COMPUTER INTERFACE

The incorporation of Retrieval-as-Learning method enhances the performance of both the Baseline and Agent S models, with a notably greater impact observed for Agent S, as shown in Table 6.

Table 6: The detailed result of ACI ablation study on $test_{sub}$ of OSWorld. The backbone model of baseline and Agent S is GPT-4o.

| Method | Success Rate (%) ↑ | | | | | |
|---|---|---|---|---|---|---|
| | OS (6) | Office (17) | Daily (16) | Profess. (10) | Workflow (16) | Overall (65) |
| Baseline | 33.33 | 5.88 | 12.50 | 10.00 | 6.25 | 10.77 |
| + Retrieval | 00.00 | 00.00 | 25.00 | 30.00 | 6.25 | 12.31 |
| Agent S (ACI-only) | 16.60 | 5.88 | 18.75 | 20.00 | 6.25 | 12.31 |
| + Retrieval | 33.33 | 11.76 | 31.25 | 30.00 | 6.25 | 20.00 |

## B  DETAILED RESULTS ON OSWORLD AND WINDOWSARENA

Table 7: Detailed success rates of baseline and Agent S using GPT-4o on OSWorld, divided by apps (domains): OS, LibreOffice Calc, LibreOffice Impress, LibreOffice Writer, Chrome, VLC Player, Thunderbird, VS Code, GIMP and Workflow involving with multiple apps.

| Method | Success Rate (%) ↑ | | | | | | | | | |
|---|---|---|---|---|---|---|---|---|---|---|
| | OS | Calc | Impress | Writer | VLC | TB | Chrome | VSC | GIMP | Workflow |
| Baseline | 1.67 | 4.26 | 6.81 | 8.70 | 9.50 | 6.67 | 15.22 | 30.43 | 0.00 | 7.46 |
| Agent S | 45.84 | 2.13 | 15.34 | 30.42 | 30.06 | 40.00 | 21.74 | 52.17 | 23.08 | 10.53 |

Table 8: Detailed success rates of Agent S using GPT-4o on WindowArena, divided by apps (domains): Chrome, Microsoft Edge, VS Code, Notepad, LibreOffice Calc, Settings, Windows Calc, Clock, VS Code, Microsoft Paint, File Explorer, LibreOffice Writer, VLC Player.

| Method | Success Rate (%) ↑ | | | | | | | | | | | |
|---|---|---|---|---|---|---|---|---|---|---|---|---|
| | Chrome | Msedge | VSC | Notepad | Lib_Calc | Settings | Win_Calc | Clock | Paint | File | Writer | VLC |
| Agent S | 17.65 | 7.69 | 29.17 | 0.00 | 0.00 | 80.00 | 0.00 | 50.00 | 0.00 | 36.84 | 0.00 | 19.05 |

## C  EXPERIENCE-AUGMENTED HIERARCHICAL PLANNING

**Observation-Aware Query**  The Manager formulates a query $Q$ based on the user task $T_u$ and initial observation $O_0$:

$$Q = LLM(T_u, O_0)$$

**Narrative Memory – Storing Full Task Experiences**  The narrative memory is indexed using an observation-aware query $Q$ formulated by the Manager. It is represented as:

$$M_n(Q) = \text{Save}(M_n, Q)$$

where $M_n$ represents the narrative memory, and $Q$ is the query generated based on the user task and initial observation $O_0$.

**Episodic Memory – Storing Successful Subtask Experiences**  The episodic memory is used by Workers to execute subtasks and is indexed using the full User Task $T_u$, subtask $s_i$, and contextual information $C_{s_i}$:

$$M_e(T_u, s_i, C_{s_i}) = \text{Save}(M_e, \langle T_u, s_i, C_{s_i} \rangle)$$

Where $M_e$ represents the episodic memory.

### C.1 MANAGER: FUSING EXTERNAL KNOWLEDGE AND INTERNAL EXPERIENCE FOR PLANNING

**External Knowledge Retrieval**  The query $Q$ is used to retrieve external knowledge $K_{\text{ext}}$ using the Perplexica search engine:

$$K_{\text{ext}} = \text{Retrieve}(\text{Web}, Q)$$

**Fusion of Internal Experience and External Knowledge**  The internal narrative memory experience $M_n(Q)$ and external knowledge $K_{\text{ext}}$ are combined using the Experience Context Fusion module:

$$K_{\text{fused}} = \text{MLLM}(M_n(Q), K_{\text{ext}})$$

**Subtask Planning**  The fused knowledge $K_{\text{fused}}$ is used by the Manager to generate a queue of subtasks $\langle s_0, s_1, \ldots, s_n \rangle$ and associated contexts $\langle C_{s_0}, C_{s_1}, \ldots, C_{s_n} \rangle$:

$$\{\langle s_0, C_{s_0}\rangle, \langle s_1, C_{s_1}\rangle, \ldots, \langle s_n, C_{s_n}\rangle\} = \text{MLLM}(K_{\text{fused}})$$

### C.2 WORKER: LEARNING FROM SUBTASK EXPERIENCE AND TRAJECTORY REFLECTION

**Subtask Execution**  Each Worker $w_i$ retrieves subtask experience $s_i$ by querying the episodic memory $M_e$:

$$E_{s_i} = \text{Retrieve}(M_e, \langle T_u, s_i, C_{s_i}\rangle)$$

**Trajectory Reflection**  The Worker reflects on the entire episode using a Trajectory Reflector $TR_i$:

$$\text{Reflection} = TR_i(\text{trajectory})$$

This reflection helps the Worker refine its strategies.

**Action Generation**  Using the retrieved subtask experience $E_{s_i}$, the Worker generates a structured response for a grounded action $a_j$:

$$a_j = \text{MLLM}(E_{s_i}, \text{observation}, \text{Reflection})$$

**Subtask Completion**  The Worker signals the end of a subtask either through DONE or FAIL:

$$\text{status} = \begin{cases} \text{DONE}, & \text{if subtask completed successfully} \\ \text{FAIL}, & \text{if subtask fails} \end{cases}$$

### C.3 SELF-EVALUATOR: GENERATING SUMMARIZED EXPERIENCES AS TEXTUAL REWARDS

**Episodic Experience Update**  If a Worker completes a subtask, the Self-Evaluator $S$ generates an Episodic Experience $E_{e_i}$ as a summary of the strategy used:

$$R_{s_i} = S(\text{Episode}_i)$$

This experience is saved back into the episodic memory, indexed by the task $T_u$, subtask $s_i$, and contextual information $C_{s_i}$:

$$M_e \leftarrow \text{Save}(M_e, \langle T_u, s_i, C_{s_i}\rangle, r_{s_i})$$

**Narrative Experience Update**   When the entire task is completed by the Manager $G$, the Self-Evaluator generates a task completion reward $r_T$, which is saved into the narrative memory, indexed by the observation-aware query $Q$ formulated by the Manager:

$$E_{n_u} = S(G(T_u))$$
$$M_n \leftarrow \text{Save}(M_n, Q, E_{n_u})$$

## C.4   ABLATIONS ON LEARNING

The results presented in Table 9 demonstrate the critical role played by both the Continual Learning component and the Self-Evaluator in enhancing the performance of Agent S.

Table 9: The detailed result of experience-augmented hierarchical planning ablation study on $test_{sub}$ of OSWorld. The backbone model of baseline and Agent S is GPT-4o.

| Method | Success Rate (%) ↑ | | | | | |
|---|---|---|---|---|---|---|
| | OS (6) | Office (17) | Daily (16) | Profess. (10) | Workflow (16) | **Overall (65)** |
| Agent S | 50.00 | 11.76 | 37.50 | 40.00 | 12.50 | 26.15 |
| - w/o Continual Memory Update | 33.33 | 11.76 | 37.50 | 30.00 | 12.50 | 23.08 |
| - w/o Self-Evaluator | 33.33 | 5.88 | 31.25 | 20.00 | 12.50 | 18.46 |
| - w/o Self-supervised Exploration | 33.33 | 5.88 | 25.00 | 20.00 | 6.25 | 15.38 |

## D   SUPPLEMENTARY EXAMPLES FOR QUALITATIVE ANALYSIS

Here we present additional examples of successful and failed tasks as supplements to the qualitative analysis in §4.2. Furthermore, we provide a more detailed error analysis to complement §4.4.

## D.1   SUCCESS EXAMPLES

In this section, we present successful task examples from a variety of domains.

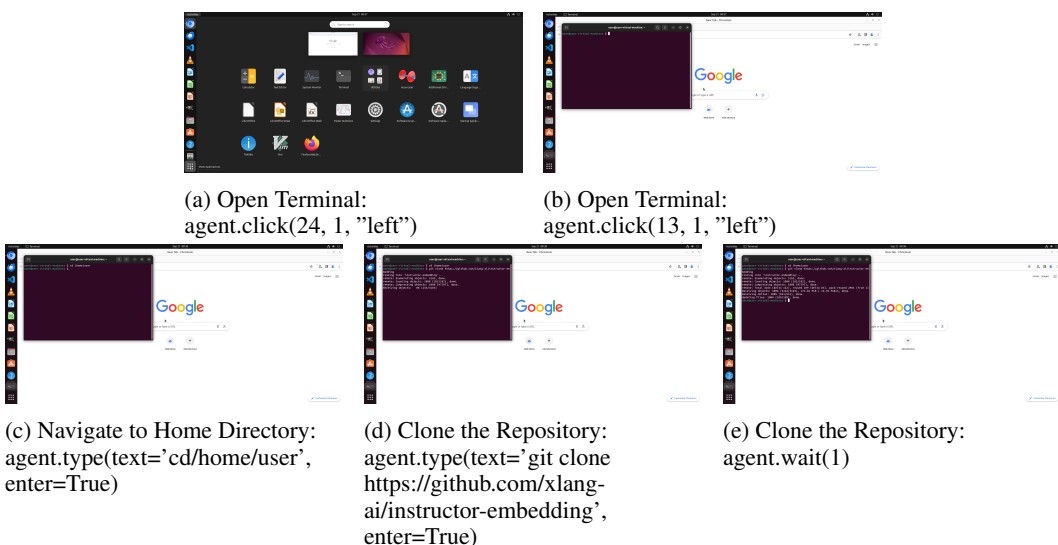

(a) Open Terminal:
agent.click(24, 1, "left")

(b) Open Terminal:
agent.click(13, 1, "left")

(c) Navigate to Home Directory:
agent.type(text='cd/home/user',
enter=True)

(d) Clone the Repository:
agent.type(text='git clone
https://github.com/xlang-
ai/instructor-embedding',
enter=True)

(e) Clone the Repository:
agent.wait(1)

Figure 9: A successful task of Multi_apps. The task instruction is: *Please help me clone the repo "https://github.com/xlang-ai/instructor-embedding" to /home/user.*. Each caption contains the plan of the subtask and its corresponding grounding action.

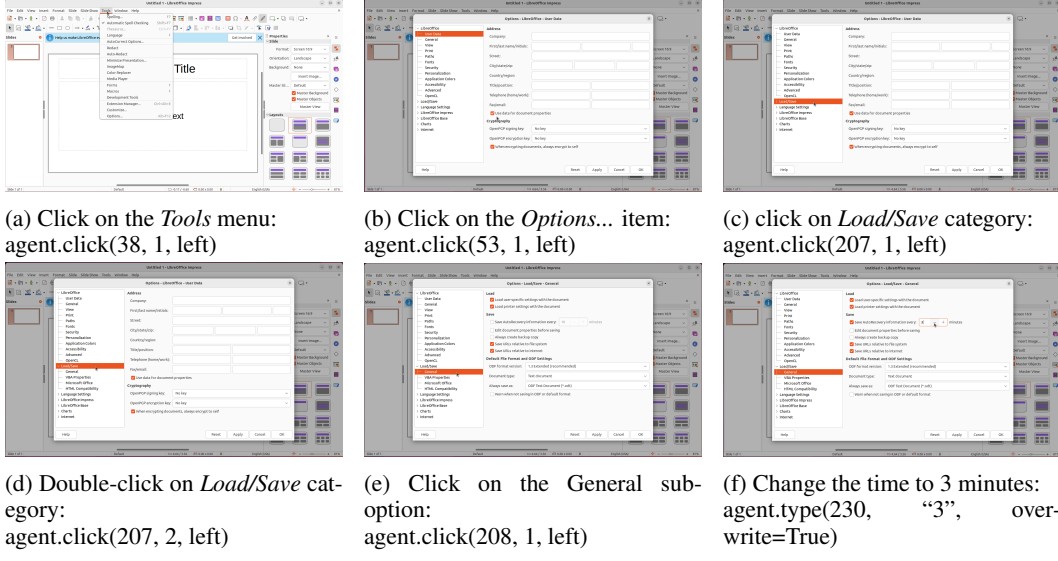

(a) Click on the *Tools* menu:
agent.click(38, 1, left)

(b) Click on the *Options...* item:
agent.click(53, 1, left)

(c) click on *Load/Save* category:
agent.click(207, 1, left)

(d) Double-click on *Load/Save* category:
agent.click(207, 2, left)

(e) Click on the General suboption:
agent.click(208, 1, left)

(f) Change the time to 3 minutes:
agent.type(230, "3", overwrite=True)

Figure 10: An example of LibreOffice Impress. The task instruction is: *Enable auto-save every 3min for me, so that I don't need to hit Ctrl-S that much.* Each caption contains the plan and its corresponding grounding action.

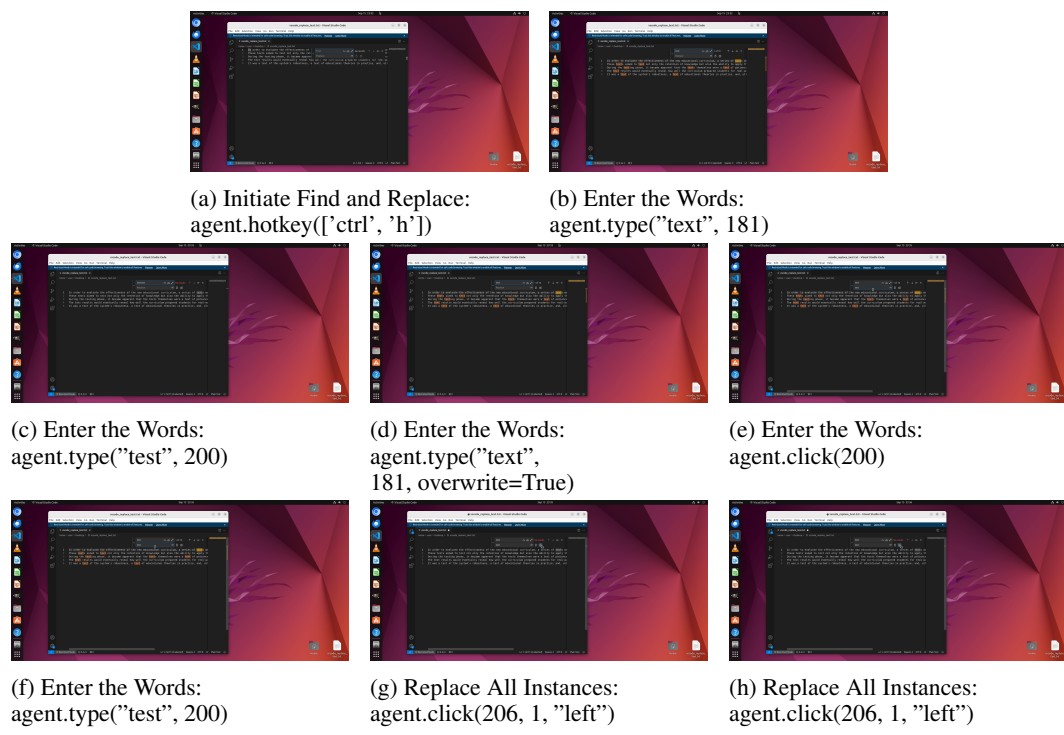

(a) Initiate Find and Replace:
agent.hotkey(['ctrl', 'h'])

(b) Enter the Words:
agent.type("text", 181)

(c) Enter the Words:
agent.type("test", 200)

(d) Enter the Words:
agent.type("text", 181, overwrite=True)

(e) Enter the Words:
agent.click(200)

(f) Enter the Words:
agent.type("test", 200)

(g) Replace All Instances:
agent.click(206, 1, "left")

(h) Replace All Instances:
agent.click(206, 1, "left")

Figure 11: A successful task of VSCode. The task instruction is: *Please help me change all the places in this document that say "text" to "test".* Each caption contains the plan of the subtask and its corresponding grounding action.

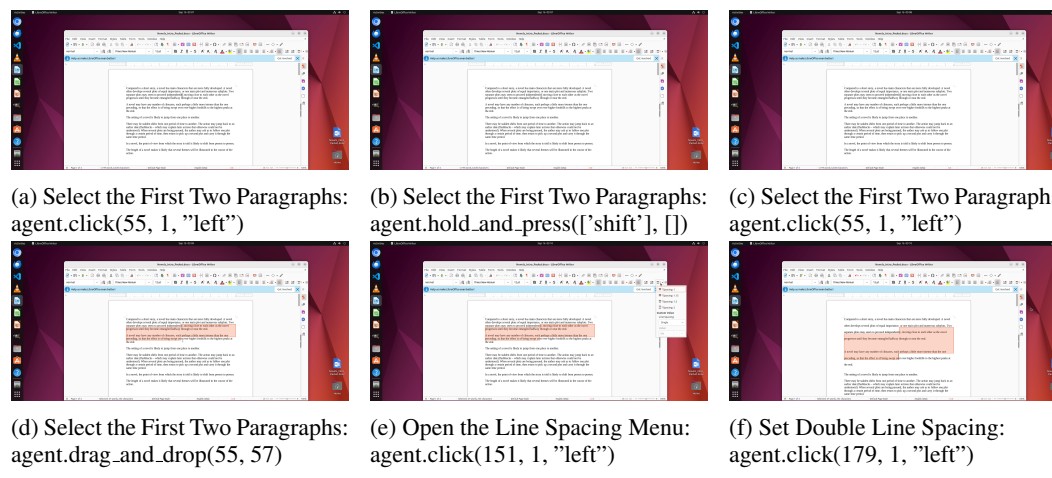

(a) Select the First Two Paragraphs: agent.click(55, 1, "left")

(b) Select the First Two Paragraphs: agent.hold_and_press(['shift'], [])

(c) Select the First Two Paragraphs: agent.click(55, 1, "left")

(d) Select the First Two Paragraphs: agent.drag_and_drop(55, 57)

(e) Open the Line Spacing Menu: agent.click(151, 1, "left")

(f) Set Double Line Spacing: agent.click(179, 1, "left")

Figure 12: A successful task of LibreOffice Writer. The task instruction is: *Make the line spacing of first two paragraph into double line spacing.* Each caption contains the plan of the subtask and its corresponding grounding action.

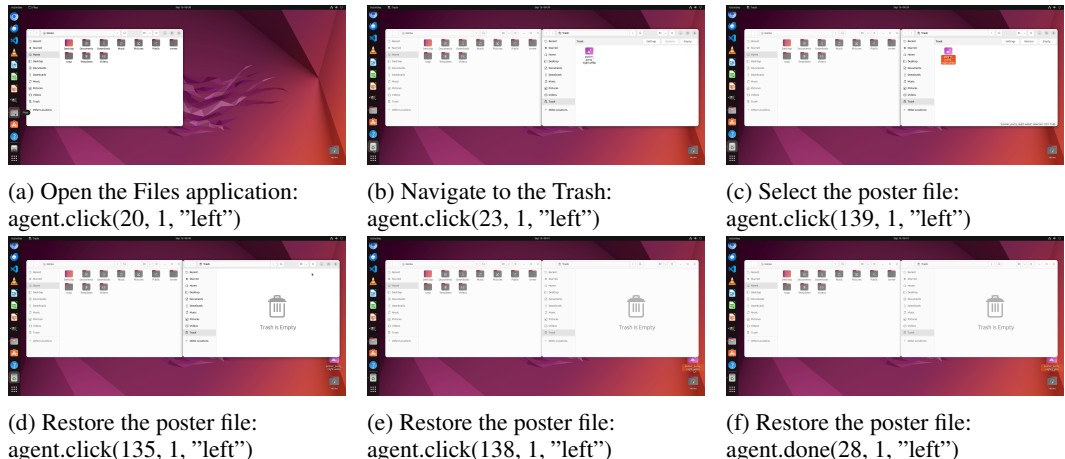

(a) Open the Files application: agent.click(20, 1, "left")

(b) Navigate to the Trash: agent.click(23, 1, "left")

(c) Select the poster file: agent.click(139, 1, "left")

(d) Restore the poster file: agent.click(135, 1, "left")

(e) Restore the poster file: agent.click(138, 1, "left")

(f) Restore the poster file: agent.done(28, 1, "left")

Figure 13: A successful task of OS. The task instruction is: *I am currently using an Ubuntu system, and I have wrongly deleted a poster of party night. Could you help me recover it from the Trash?* Each caption contains the plan of the subtask and its corresponding grounding action.

Although the agent successfully completes the tasks depicted in Figure 9 10 11 12 13, there are still issues present in its execution trajectories. For instance, during the task in Figure 11, the agent incorrectly enters the word into the wrong field at Figure 11 (c), although this mistake is corrected promptly. Furthermore, in the course of the task demonstrated in Figure 12, the agent exhibits inappropriate actions at Figure 12 (a)(b)(c). Additionally, while performing the task depicted in Figure 13, the agent fails to recognize the completion of the task at Figure 13 (d), subsequently attempting to recover an already existing file on the desktop at Figure 13 (e)(f). These issues highlight the inherent challenges in achieving consistently reliable behavior, even when tasks are nominally completed.

## D.2 DETAILED ERROR ANALYSIS AND FAILURE EXAMPLES

In this section, we analyze the sources of execution errors as defined in §4.4, followed by presenting several examples of failed tasks, each with a detailed error analysis provided for the respective case. Empirically, Grounding and planning errors often directly lead to execution errors (e.g., failing to interact with the correct target element can result in repetitive actions, and incorrect planning

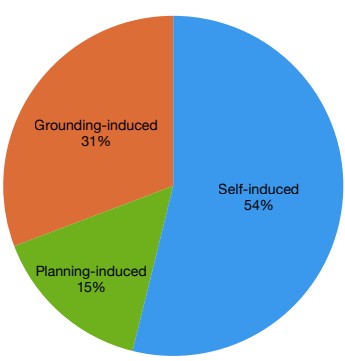

Figure 14: The error sources of the overall 39 execution errors.

messages can lead to wrong decisions while performing the task). We reviewed all 39 execution errors in errors on $test_{sub}$ of OSWorld that Agent S failed to complete, as shown in Figure 14, and found that 46% were caused by planning or grounding errors. This indicates that reducing these errors, particularly grounding errors, which frequently cause repetitive actions, could significantly improve performance.

During the task in Figure 15, the agent simultaneously makes planning, execution, and grounding errors. First, the inaccurate planning information in Figure 15 (a) suggests typing '1' instead of 'No. 1' in the cell constitutes a planning error, leading the agent to type the incorrect value. Additionally, the agent's attempt to drag the fill handle from 'B2' to 'B23' in Figure 15 (b) fails due to the selection of erroneous elements and coordinates, which can be classified as a grounding error. Furthermore, the agent continues to try to execute the subtask 'Drag the Fill Handle' with repetitive actions in Figure 15 (c)(d)(e)(f), overlooking the prior grounding error and being unable to correct its behavior timely, which is indicative of an execution error.

Another type of planning error emerges while the agent is executing the task shown in Figure 16. The plan generated by the agent is flawed, as it incorporates an irrelevant subtask "Updating of Chrome", which does not pertain to the intended goal. Additionally, the resulting subtask sequence is incorrect, as it erroneously prioritizes such subtask, as illustrated in Figure 16 (c)(d). This fundamental planning deficiency propagates into an execution error, preventing the agent from successfully turning off the extension, as demonstrated in the subsequent figures.

The failed task depicted in Figure 17 illustrates a scenario where the agent makes a grounding error, which subsequently leads to an execution error. After adding the Alpha Channel, the agent attempts to select the 'Fuzzy Select Tool' from the toolbox to target the background. However, instead of selecting the correct element (represented by the magic wand icon), the agent consistently grounds to the incorrect element, 'Activity', located at the top-left corner. This misselection brings the system to its 'Overview' state. The agent then switches back to GIMP but continues to incorrectly select 'Activity', mistakenly identifying it as the 'Fuzzy Select Tool'. This repeated incorrect action is demonstrated in Figure 17(e)(f)(g)(h). It is evident that the agent fails to correct its behavior promptly when facing this issue, which can be considered an execution error stemming directly from the initial grounding error.

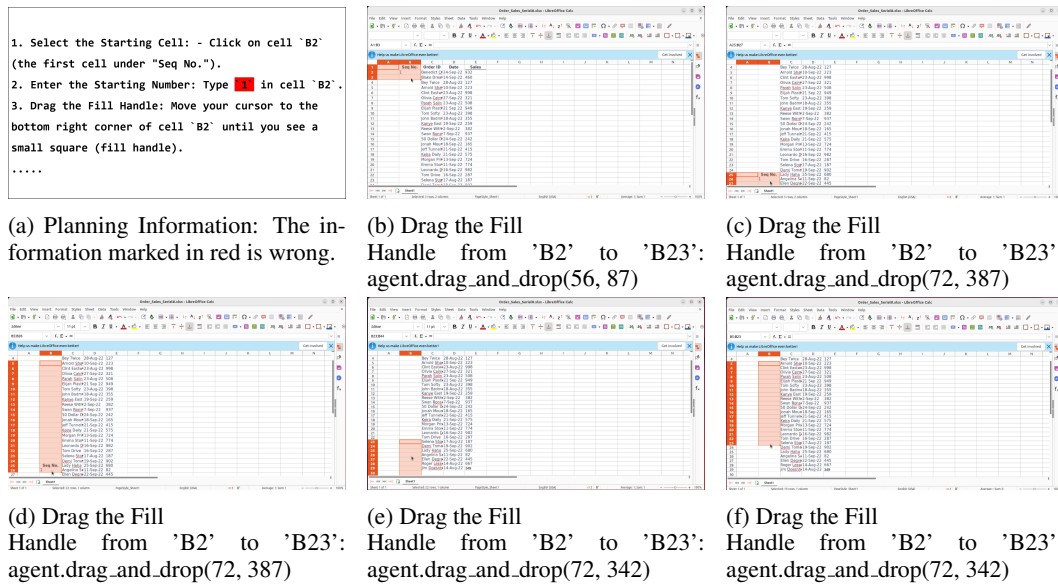

(a) Planning Information: The information marked in red is wrong.

(b) Drag the Fill Handle from 'B2' to 'B23': agent.drag_and_drop(56, 87)

(c) Drag the Fill Handle from 'B2' to 'B23': agent.drag_and_drop(72, 387)

(d) Drag the Fill Handle from 'B2' to 'B23': agent.drag_and_drop(72, 387)

(e) Drag the Fill Handle from 'B2' to 'B23': agent.drag_and_drop(72, 342)

(f) Drag the Fill Handle from 'B2' to 'B23': agent.drag_and_drop(72, 342)

Figure 15: An failed task of LibreOffice Calc. The task instruction is: *Fill the Sequence Numbers as "No. #" in the "Seq No." column*

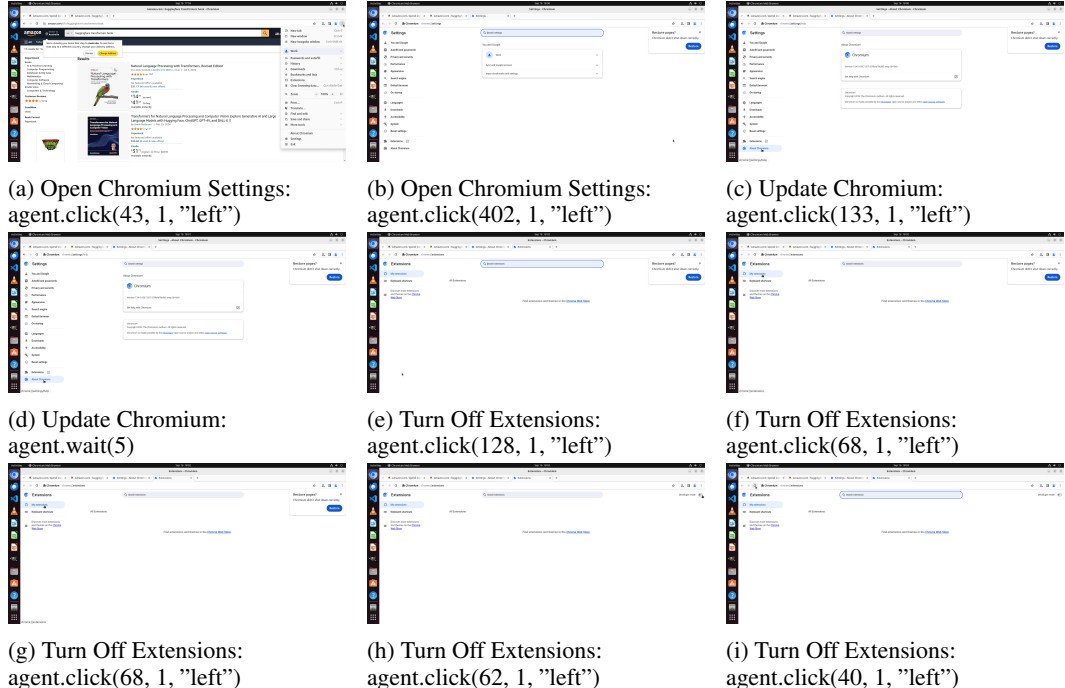

(a) Open Chromium Settings: agent.click(43, 1, "left")

(b) Open Chromium Settings: agent.click(402, 1, "left")

(c) Update Chromium: agent.click(133, 1, "left")

(d) Update Chromium: agent.wait(5)

(e) Turn Off Extensions: agent.click(128, 1, "left")

(f) Turn Off Extensions: agent.click(68, 1, "left")

(g) Turn Off Extensions: agent.click(68, 1, "left")

(h) Turn Off Extensions: agent.click(62, 1, "left")

(i) Turn Off Extensions: agent.click(40, 1, "left")

Figure 16: An failed task of Chrome. The task instruction is: *Can you help me clean up my computer by getting rid of all the tracking things that Amazon might have saved? I want to make sure my browsing is private and those sites don't remember me.* Each caption contains the plan of the subtask and its corresponding grounding action.

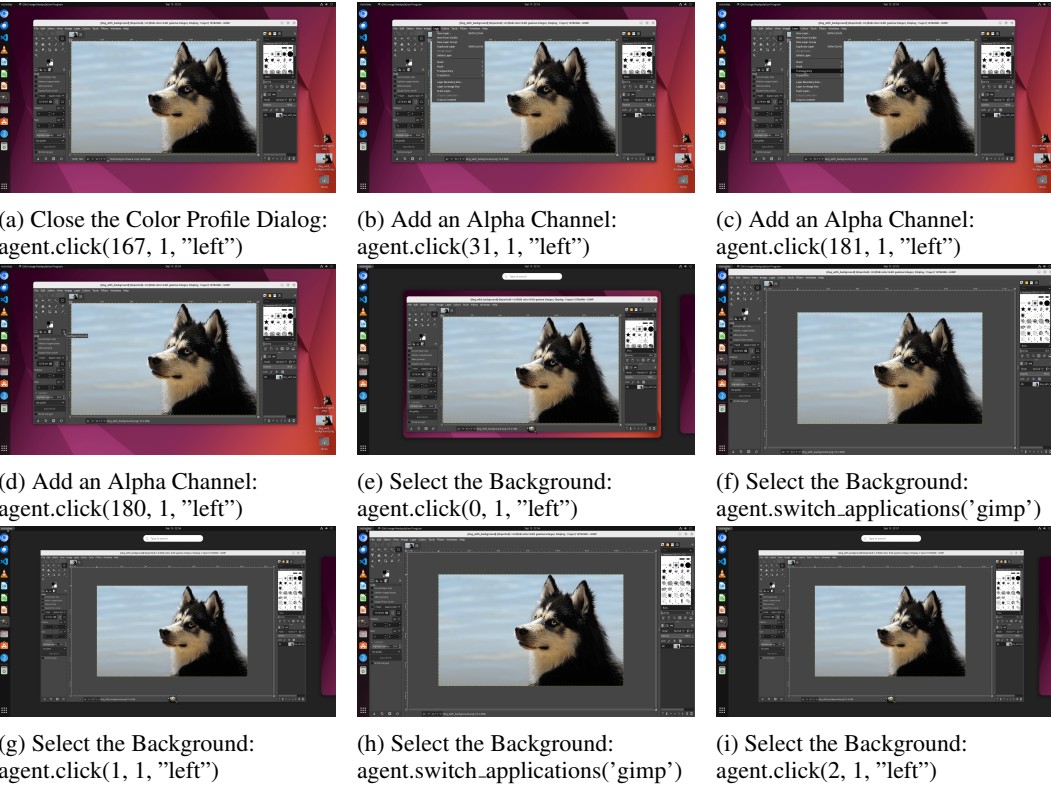

(a) Close the Color Profile Dialog: agent.click(167, 1, "left")

(b) Add an Alpha Channel: agent.click(31, 1, "left")

(c) Add an Alpha Channel: agent.click(181, 1, "left")

(d) Add an Alpha Channel: agent.click(180, 1, "left")

(e) Select the Background: agent.click(0, 1, "left")

(f) Select the Background: agent.switch_applications('gimp')

(g) Select the Background: agent.click(1, 1, "left")

(h) Select the Background: agent.switch_applications('gimp')

(i) Select the Background: agent.click(2, 1, "left")

Figure 17: An failed task of GIMP. The task instruction is: *Could you make the background of this image transparent for me?* Each caption contains the plan of the subtask and its corresponding grounding action.

