# OpenReview forum: "Agent S: An Open Agentic Framework that Uses Computers Like a Human"
_ICLR.cc/2025/Conference — ICLR 2025 Poster_

### Official Review · Reviewer_7E6Z · 2024-10-28

**Soundness:** 3
**Presentation:** 2
**Contribution:** 2
**Rating:** 5
**Confidence:** 3

**Summary:**

This paper introduces a new multimodal large language model agent for GUI control, called Agent S. Its main feature is the ability to complete various tasks in a GUI interface through direct keyboard and mouse control. Comparing with other agents, Agent S incorporates an experience-augmented hierarchical planning method, which enhances the agent's ability to decompose tasks based on trajectories stored in memory. Experiments demonstrate that this framework is versatile, capable of executing various GUI-oriented tasks across different systems(Ubuntu and Windows).

**Strengths:**

The advantages of this paper are that it has a clear line of thought, smooth transitions, and is easy to follow. The main architecture diagram of the agent is concise and clear, allowing for a straightforward understanding of the information flow. The selection of experimental environments is well-considered, taking into account both Ubuntu and Windows systems, which proves the effectiveness and generalization of the framework.

**Weaknesses:**

However, the experiments in this paper are not sufficiently comprehensive. First, the baselines used are not enough, and in the comparison results of the main experiments, MLLMs are used simply instead of MLLM agents. The experimental results based on single MLLMs that only input images and accessibility trees are not convincing enough. Existing agents such as Cradle and Claude 3 perform well using only keyboard and mouse inputs without requiring additional accessibility trees. As a result, the third contribution of this paper also limits the applicability of Agent S and raises doubts about whether all GUIs provide accessibility tree inputs and whether such inputs are necessary. Thus, the contributions of this paper may seem insufficiently innovative.

reference：
[1]Tan W, Zhang W, Xu X, et al. Cradle: Empowering Foundation Agents towards General Computer Control[C]//NeurIPS 2024 Workshop on Open-World Agents.

**Questions:**

First, I have some doubts about the generalizability of accessibility trees. Do all GUIs have accessibility trees, and do systems like Ubuntu and Windows have similar forms of accessibility trees? I hope the article can clarify these points. Secondly, I would like the article to compare Agent S with different agents, evaluating whether Agent S demonstrates better experimental results compared to existing GUI agents. Lastly, the ablation study does not adequately cover all points of contribution. An experiment should clarify the performance of Agent S when solely relying on image input without ACI.

---

> ### Comment · Reviewer_7E6Z · 2024-11-15
> **Additional Explanation of My Concerns**
>
> Thanks for feedback by Associate Program Chairs. The two suggestions are actually the same issue, and I will restate my concerns here.
>
> Agent S is a system agent based on MLLM, with various additional modules such as memory, evaluator, etc. to assist MLLM in making decisions. It is not enough for the baseline to use only pure GPT4o, claude-3 and other models. The same agent system should be used for comparison to increase the persuasiveness of the experimental results.
>
> Cradle[1] and claude 3[2] are the methods I know that can be used as baselines, so they need to be compared. Cradle is also an MLLM agent with GPT4o to control mouse and keyboard. Claude-3 is emphasized because Osworld itself uses claude-3 as a baseline. Agent s should compare the difference between agent s w/Claude -3 and pure Claude - 3, rather than comparing the difference between agent s w/Claude -3.5 and Claude - 3.
>
> Reference:
> [1]Tan W, Zhang W, Xu X, et al. Cradle: Empowering Foundation Agents towards General Computer Control[C]//NeurIPS 2024 Workshop on Open-World Agents.
> [2]Anthropic. The claude 3 model family: Opus, sonnet, haiku. 2024. URL https://api.semanticscholar.org/CorpusID:268232499.

---

> ### Author Response · Authors · 2024-11-22
> **Thank you for the review!**
>
> We thank reviewer **7E6Z** for their detailed review of our work. We would like to take the opportunity to answer their questions and respond to their comments about our work.
>
> `W1: MLLMs are used with MMAgent in the Baselines`
>
> The baselines in Table 1 are not just MLLMs with tree and image inputs but rather MLLMs used within the MMAgent framework (Xie et al.), which incorporates working memory management, accessibility tree filtering, detailed prompts, parsing, and stepwise code execution. MMAgent achieved state-of-the-art results on the OSWorld benchmark at the time of our submission. We thank reviewer 7E6Z for highlighting this, and we have updated the presentation to clarify that the baselines include the MMAgent framework.
>
> `W2: Additional comparisons with existing agents`
>
> | Agent                                | Modality                         | Success Rate (%) |
> |--------------------------------------|----------------------------------|------------------|
> | MMAgent w. Claude-3-opus [2]         | Screenshot                      | 2.42             |
> | MMAgent w. Claude-3-opus [2]         | Screenshot + Accessibility Tree | 4.41             |
> | Cradle [1]                           | Screenshot                      | 7.81             |
> | OpenInterpreter* w. GPT-4o [3]       | Screenshot + Accessibility Tree | 8.95             |
> | Friday* w. GPT-4o [4]                | Screenshot + Accessibility Tree | 11.11            |
> | MMAgent w. GPT-4o [2]                | Screenshot + Accessibility Tree | 11.21            |
> | MMAgent w. GPT-4V [2]                | Screenshot + Accessibility Tree | 12.17            |
> | Agent S w. GPT-4o                    | Screenshot + Accessibility Tree | 20.58            |
>
>
> Here, we report the results of OpenInterpreter[3], FRIDAY[4], and CRADLE[1] as other agentic frameworks that are suitable for the OSWorld benchmark. Scores for OpenInterpreter and Friday (represented by *) were evaluated by a third party. Cradle and MMAgent scores are as represented in the respective papers. We have also updated these scores in Table 1 of the paper.
> The CRADLE agent [1] achieved a 7.81% success rate on the OSWorld benchmark, which is much lower than our best result of 20.58%. While vision-only agents can potentially be powerful, the visual grounding abilities of current MLLM models are not sophisticated enough to handle precise grounding for UI manipulation [2, 5] The Claude-3-opus model with MMAgent [2] gets a 4.41% score in Screenshot + Accessibility Tree input and only 2.42% in Screenshot-only mode, reiterating the limitations of current MLLMs at reasonable visual grounding in UIs.
>
> `Q1: Do all GUIs have accessibility trees?`
>
> Accessibility trees are widely supported across all major OSs, including Windows (via UI Automation), macOS (through Apple's Accessibility API), Linux (using AT-SPI), as well as Android and iOS with their respective Accessibility APIs. Browsers like Chrome, Safari, Firefox, etc. all provide accessibility trees based on the Document Object Model. These accessibility frameworks are integral to modern OS design, driven by regulatory requirements such as the EAA[6], and influenced by industry standards like WCAG[7]. Frameworks like Selenium and Appium use accessibility trees for GUI testing and automation. Commercial vendors like UiPath and Automation Anywhere also use accessibility trees for Robotic Process Automation for GUIs. In general, accessibility trees are a common approach for GUI automation and are supported by all OSs.
>
> `Q1: Systems like Ubuntu and Windows have similar forms of accessibility trees`
>
> Ubuntu (AT-SPI), Windows (UIA) and MacOS (AXTrees) all represent the UI as a tree structure, starting from a root element (such as the desktop or main application window) and branching out to various UI components like windows, menus, and controls. In each framework, UI components are represented as accessible objects with properties such as role, name, and state. In our experiments with Windows and Ubuntu, we observed that the main differences came from the nomenclature used across the different frameworks which is quite easy to adapt from the respective documentation. We have added examples of Accessibility trees from Windows and Ubuntu in XML format to the Supplementary material for a better idea.

---

> > ### Author Response · Authors · 2024-11-22
> > **Addressing further concerns**
> >
> > `Q2: Clarification on ablation study with Agent S relying only on Screenshot`
> >
> > Agent S requires both an accessibility tree and a screenshot for operation. We use the accessibility tree to allow the agent to select and ground specific UI elements. The ability of the SOTA Foundation Models (GPT-4o, Claude-3.5-sonnet, etc.) to precisely ground to arbitrary points within an image or a UI is extremely limited [2, 5]. This is evidenced by the low scores obtained by CRADLE [1] and Vision-only results from MMAgent [2] We use screenshots as reasoning and verification input, allowing the agent to observe the impact of its actions. We compare our agent with the best-performing baseline agent (MMAgent w. GPT-4V), which uses both Screenshot and Accessibility Tree, thus providing a fair comparison.
> >
> > `W2: Comparison using claude-3-opus and claude-3.5-sonnet`
> >
> > The Claude-3-opus has very restrictive rate limits and 5x the cost of Claude-3-5-sonnet-20240620, making it highly expensive (time and cost) for us to run the entire test set of OSWorld. We ran evaluations of both models on the smaller subset, which we used to conduct ablation studies ((65 examples, results shown in the table below), showing the improvement brought about by AgentS as well as the Baseline MMAgent performances with both models.
> >
> > |                     | Claude-3-opus-20240229 | Claude-3.5-sonnet-20240620 |
> > |---------------------|-------------------------|----------------------------|
> > | **MMAgent**         | 7.69%                  | 9.24%                     |
> > | **Agent S**         | 12.18%                 | 22.95%                    |
> >
> > [1] Tan W, Zhang W, Xu X, et al. Cradle: Empowering Foundation Agents towards General Computer Control[C]//NeurIPS 2024 Workshop on Open-World Agents.
> >
> > [2] Tianbao Xie, et al.. Osworld: Benchmarking multimodal agents for open-ended tasks in real computer environments. CoRR, abs/2404.07972, 2024. doi:10.48550/ARXIV.2404.07972.
> >
> > [3] open-interpreter, 2024. URL https://github.com/OpenInterpreter/open-interpreter.
> >
> > [4] Zhiyong Wu, Chengcheng Han, Zichen Ding, Zhenmin Weng, Zhoumianze Liu, Shunyu Yao, Tao Yu, and Lingpeng Kong. Os-copilot: Towards generalist computer agents with self-improvement, 2024.
> >
> > [5] Boyuan Zheng, Boyu Gou, Jihyung Kil, Huan Sun, and Yu Su. Gpt-4v(ision) is a generalist web agent, if grounded. In Forty-first International Conference on Machine Learning, ICML 2024, Vienna, Austria, July 21-27, 2024. OpenReview.net, 2024a. URL https://openreview.net/forum?id=piecKJ2DlB.
> >
> > [6] https://eur-lex.europa.eu/legal-content/EN/TXT/?uri=CELEX%3A32019L0882
> >
> > [7] https://www.w3.org/WAI/standards-guidelines/wcag/

---

> ### Author Response · Authors · 2024-11-25
> **Looking forward to further discussion**
>
> We thank you for your detailed review of our work. We appreciate your acknowledgment of our work as following a clear line of thought and having smooth transitions, as well as our experiments being conducted in well-considered environments that demonstrate the effectiveness and generalization of the framework.
>
> We are sincerely grateful for the effort you have invested in helping us improve our work. Your feedback on our work has been invaluable, and we have carefully addressed your concerns in our response. With the discussion period deadline approaching, we would greatly appreciate any further comments you might have!

---

> > ### Author Response · Authors · 2024-11-27
> >
> > Dear Reviewer 7E6Z,
> >
> > As today is the final day for uploading a revision, we wanted to kindly follow up regarding our responses to your comments on our submission.
> >
> > If there are any additional points or suggestions you’d like us to address, we would be happy to incorporate them before the deadline. We would also greatly appreciate it if you could let us know whether your concerns or questions have been fully addressed.
> >
> > Thank you once again for your thoughtful feedback and support!
> >
> > Best regards,
> > Authors

---

### Official Review · Reviewer_KnHm · 2024-10-31

**Soundness:** 3
**Presentation:** 3
**Contribution:** 3
**Rating:** 8
**Confidence:** 5

**Summary:**

This paper presents Agent S, a novel framework for GUI-based operating system control that integrates three key strategies: experience-augmented hierarchical planning, continual memory update, and an Agent-Computer Interface (ACI). The framework introduces an effective memory mechanism with initialization and continuous update algorithms, demonstrating state-of-the-art performance on computer use benchmarks like OSWorld. A notable contribution is the carefully designed ACI that addresses the unique challenges of MLLM agents interacting with desktop environments.

**Strengths:**

1. **Novel and Effective Memory Mechanism**: Introduces a well-designed memory system with both narrative and episodic components
Provides clear algorithms for both initial memory construction and continuous updates
Demonstrates a complete closed-loop system with practical effectiveness


2. **Insightful Analysis of Agent-Computer Interaction**: Deep analysis of fundamental challenges in MLLM-based computer control
Identifies key issues like discrete time response, lack of internal coordinate systems, and inefficient feedback processing
Addresses the limitations of traditional API/script-based automation approaches


3. **Innovative ACI Design**: Proposes a dual-input strategy combining visual and accessibility tree information
Implements bounded action space with concurrent feedback
Successfully bridges the gap between MLLM agents and GUI control requirements


4. **Strong Empirical Results**: Achieves SOTA performance on established benchmarks
Provides comprehensive experimental validation

**Weaknesses:**

1. **Limited Problem Definition**: The paper could benefit from a more detailed introduction to computer automation tasks
Key concepts like planning, execution, and grounding could be better explained for readers new to the field


2. **Presentation Issues**: Some overlap between Figures 3 and 4 that could be consolidated or better differentiated
Technical details of the ACI implementation could be more thoroughly described

**Questions:**

1. **Citation Format Issues**: References to OSWorld (Xie et al., 2024) and WindowsAgentArena (Bonatti et al., 2024) need clarification as they appear to be forward citations


2. **Figure Organization**: Have the authors considered combining Figures 3 and 4 for better presentation, or focusing Figure 4 more specifically on memory aspects?

---

> ### Author Response · Authors · 2024-11-22
> **Thank you for the review!**
>
> We thank reviewer **KnHM** for their review and insightful comments. We have updated our paper based on their points (marked in red). We address their concerns as follows:
>
> `W1: Limited Problem Definition`
>
> We have added descriptions motivating the GUI automation problem as a Partially Observable Markov Decision Process in Section 3. We have also added details about Hierarchical Planning and its general Workflow to clarify planning and execution in Section 3.. Finally we have introduced Grounding to UI Elements in Section 3.3 to give readers a better background about the problem and our method. We thank the reviewer for bringing this presentation deficiency to our notice.
>
> `W2 and Q2: Presentation Issues and Figure Organization:`
>
> While we did consider combining figures 3 and 4, we decided to separate them out as Figure 3 is a general overview of the framework, while Figure 4 describes the learning process - self-supervised exploration and continual learning. We have simplified Figure 4 to explicitly refer only to the learning and memory constructions and clearly delineated the two stages. We have also expanded Appendix A. with an illustration describing the overview of our Agent Computer Interface.
>
> `Q1: Citation Format Issues`
>
> The OSWorld paper (Xie et al., 2024) is accepted at NeurIPS 2024, resulting in the forward citation. In our bibliography, we have used the DBLP entry corresponding to this paper. The WindowsAgentArena paper (Bonatti et al., 2024) is a concurrent work available as a preprint. This work was highly suitable for testing the generalization of our framework and allowed us to demonstrate quick and easy adaptation of our method to a new setting. We have also updated the entry for this paper from DBLP in our updated manuscript.

---

> ### Author Response · Authors · 2024-11-25
> **Looking forward to further discussion**
>
> We thank you for your insightful comments and your positive comments about our framework as having a Novel and Effective Memory Mechanism, Innovative ACI design, showing Strong Empirical Results, and providing an Insightful analysis of Agent Computer Interaction. We have taken your points in the review into consideration and updated our manuscript accordingly.
>
> We are sincerely grateful for the effort you have invested in helping us improve our work. Your feedback on our work has been invaluable, and we have carefully addressed your concerns in our response. With the discussion period deadline approaching, we would greatly appreciate any further comments you might have!

---

> > ### Author Response · Authors · 2024-11-27
> >
> > Dear Reviewer KnHm,
> >
> > As today is the final day for uploading a revision, we wanted to kindly follow up regarding our responses to your comments on our submission.
> >
> > If there are any additional points or suggestions you’d like us to address, we would be happy to incorporate them before the deadline. We would also greatly appreciate it if you could let us know whether your concerns or questions have been fully addressed.
> >
> > Thank you once again for your thoughtful feedback and support!
> >
> > Best regards,
> > Authors

---

### Official Review · Reviewer_YYG1 · 2024-11-01

**Soundness:** 3
**Presentation:** 3
**Contribution:** 1
**Rating:** 5
**Confidence:** 4

**Summary:**

The paper proposes Agent S, a computer agent framework which can directly interact with computers through GUI interface. The proposed framework mainly consists of three components, a manager to manage external and internal memory for subtasks planning, a worker to complete subtasks with an episodic memory trajectory reflector, and a self-evaluator to summarize task experiences. The authors also propose an agent-computer interface as an abstraction layer. The proposed framework is evaluated on OSWorld and seen an increase of 9.37% success rate.

**Strengths:**

1. The performance of the proposed framework on OSWorld benchmark is quite good.
2. The proposed framework is well-engineered and the evaluation is systematic.
3. The presentation and visualization of the paper is good.

**Weaknesses:**

1. It would be unfair to compare the framework only to the baseline from OSWorld, which is a benchmark paper, not a methodology paper.
2. The self-supervised exploration process is not realistic in actual deployments and I believe it will lead to overfitting.
3. The ACI proposed in this paper can only act on selected elements in the accessibility tree, which somewhat sacrifices flexibility for performance because you cannot click on every coordinate of the screen.
4. While the framework is well designed, it doesn’t introduce much new. For example, it’s quite obvious that doing some early exploration, using external resources, and OCR would help. In addition, these processes, as well as subtask planning and self-evaluation would significantly slow down the task, which is already quite slow, and would cost more OpenAI tokens.

**Questions:**

1. What is the average time cost to complete a task using OpenAI API-based models and self-hosted models?
2. Does the continued memory growth consume more LLM input context tokens? What is the average context length?

---

> ### Author Response · Authors · 2024-11-22
> **Thank you for the review!**
>
> We thank the reviewer **YYG1** for their valuable comments. We address their comments and concerns as follows:
>
> `W1: Fairness of comparison`
>
> While OSWorld is a benchmark paper, it also presents the MMAgent, the state-of-the-art result on this benchmark at the time of submission of this paper. We thank the reviewer for bringing up the point about the lack of other baseline evaluations in our presentation. We have added the following agent baselines to our paper as they are suitable agents for the OSWorld environment:
>
> | Agent                                | Modality                         | Success Rate (%) |
> |--------------------------------------|----------------------------------|------------------|
> | MMAgent w. Claude-3-opus [2]         | Screenshot                      | 2.42             |
> | MMAgent w. Claude-3-opus [2]         | Screenshot + Accessibility Tree | 4.41             |
> | Cradle [1]                           | Screenshot                      | 7.81             |
> | OpenInterpreter* w. GPT-4o [3]       | Screenshot + Accessibility Tree | 8.95             |
> | Friday* w. GPT-4o [4]                | Screenshot + Accessibility Tree | 11.11            |
> | MMAgent w. GPT-4o [2]                | Screenshot + Accessibility Tree | 11.21            |
> | MMAgent w. GPT-4V [2]                | Screenshot + Accessibility Tree | 12.17            |
> | Agent S w. GPT-4o                    | Screenshot + Accessibility Tree | 20.58            |
>
> `W2: Practicality of the self-supervised exploration process`
>
> Our Self-supervised exploration process does not need any human annotation and is performed on new, LLM-generated tasks **in an offline phase prior to actual deployment on test tasks** (which can be considered as the “training” phase of our in-context learning method). We do not perform any exploration on tasks from the test set. Currently, we use 427 tasks to explore; however, their number can be scaled up, and they can be extended to a large number of domains, applications, websites, and OSs to enlarge the knowledge base. This exploration allows our Agent to better understand the environment and interfaces without overfitting to specific tasks.
> In addition to using the experience gained from self-supervised exploration, we also utilize web knowledge for planning. If the retrieved experience is not helpful to the task, we rely on web search. We use an MLLM as a Knowledge Fusion module to judge whether the retrieved experience and search results are helpful and reliable, then combine them into integrated knowledge. Through this module, Agent S generates an end-to-end plan that effectively utilizes the most appropriate information without over-reliance on either source.
>
> `W3: The ACI proposed in this paper can only act on selected elements in the accessibility tree`
>
> While our ACI limits the agent's interaction with elements, this tradeoff between precision vs. flexibility works highly in favor of our Agent. Current MLLM models struggle with coordinate-level grounding, resulting in a very high grounding error [1]. By ensuring more precision of grounding, we can improve agents along the dimensions of planning and execution while visual grounding models catch up. Furthermore, our ACI can be easily extended to include more general instructions that can directly refer to coordinates instead of elements in the future.  Our ablations study (Figure 6) also shows that ACI allows more effective learning from experience.
>
> `W4: Novelty of Agent S framework`
>
> Agent-S is a new agentic framework that integrates experience-augmented hierarchical planning, self-supervised continual memory update, and an ACI for MLLM-based GUI agents to perform complex computer tasks at a keyboard-mouse level.
> 1. Self-supervised exploration is an important contribution as it shows that agents can explore without ground-truth labels or training sets to better understand the environment. Since the exploration can be performed in an offline phase prior to actual deployment, it is possible to scale up the coverage of this exploration.
> 2. The addition of direct web knowledge is both novel and useful, as it can be easily accessed, gets frequent updates, can be directly utilized by GUI agents at runtime, and has been missing in previous GUI automation works.
> 3. Both self-supervised exploration and web knowledge retrieval synergize well with hierarchical planning to decompose complex tasks into executable subtasks, which is a novel finding.
> 4. The design of ACIs for GUI agents allows MLLM-based agents to operate computers more precisely and focus on improving planning and execution without worrying about grounding and missing feedback. We also show that an ACI is highly effective at improving the ability of GUI agents to learn from experience (Figure 6).
>
> [1] Tianbao Xie, et al. Osworld: Benchmarking multimodal agents for open-ended tasks in real computer environments. CoRR, abs/2404.07972, 2024. doi:10.48550/ARXIV.2404.07972.

---

> ### Author Response · Authors · 2024-11-22
> **Addressing questions**
>
> `W4 and Q1: Average time cost for Openai and Self-hosted models`
>
> We calculated the average time cost per task for running OpenAI models: GPT-4o (428.4 seconds), GPT-4o-mini (308.3 seconds), and self-hosted model: Llama-3.2-90B (366.9 seconds.) This average is calculated on a subset of 65 tasks. We would like to clarify that the performance of current GUI agents lags significantly behind human performance. In this work, we focus on the general task of improving the ability of agents to successfully complete tasks over the more engineering-oriented task of optimizing the time-cost. Consequently, Agent-S nearly doubles the performance of the baseline MMAgent. Future work could focus on optimizing the framework by reducing the number of MLLM modules, using KV cache optimization, and fine-tuning smaller MLLMs for GUI tasks.
>
> `Q2: Continued memory growth does not expand the context length`
>
> We do not keep the agent's experience in the LLM’s context window; rather, we store it separately in a database. Relevant experiences are retrieved from the memory based on a query formulated using the current observation and the task instruction. The retrieved experience does not include the full trajectory of the experience but rather a concise summary. Thus, the continued memory growth does not impact the Agent's input context length. Our Worker, which retrieves subtask level experience, has an average context size of 19184.6 tokens, and Manager, which retrieves task level experience, has an average context size of 4,719.7 tokens. This size depends only on the observations and does not increase with memory growth.

---

> ### Author Response · Authors · 2024-11-25
> **Looking forward to further discussion**
>
> We thank you for your valuable comments and appreciate the acknowledgment of our framework as showing good performance on OSWorld, demonstrating systematic evaluation, and having good presentation and visualization. We have addressed your concerns about our work in detail and provided additional results to answer your questions.
>
> We are sincerely grateful for the effort you have invested in helping us improve our work. Your feedback on our work has been invaluable, and we have carefully addressed your concerns in our response. With the discussion period deadline approaching, we would greatly appreciate any further comments you might have!

---

> > ### Author Response · Authors · 2024-11-27
> >
> > Dear Reviewer YYG1,
> >
> > As today is the final day for uploading a revision, we wanted to kindly follow up regarding our responses to your comments on our submission.
> >
> > If there are any additional points or suggestions you’d like us to address, we would be happy to incorporate them before the deadline. We would also greatly appreciate it if you could let us know whether your concerns or questions have been fully addressed.
> >
> > Thank you once again for your thoughtful feedback and support!
> >
> > Best regards, Authors

---

### Official Review · Reviewer_meL5 · 2024-11-03

**Soundness:** 3
**Presentation:** 3
**Contribution:** 3
**Rating:** 6
**Confidence:** 3

**Summary:**

The paper introduces Agent S, a groundbreaking framework designed to automate complex, multi-step tasks on computers through Graphical User Interface (GUI) interaction for human usage. Agent S tackles the challenges of acquiring domain-specific knowledge, planning over long task horizons, and navigating dynamic interfaces by employing experience-augmented hierarchical planning, which leverages both external web knowledge and internal experience retrieval. It also utilizes an Agent-Computer Interface (ACI) to enhance the reasoning and control capabilities of GUI agents based on Multimodal Large Language Models (MLLMs). The framework demonstrated significant improvements in task success rates on the OSWorld benchmark and showed broad generalizability to different operating systems, setting a new state-of-the-art in autonomous GUI agent performance.

**Strengths:**

1. Agent S stands out for its task automation through experience-augmented hierarchical planning. This method harnesses external web knowledge and draws upon internal memories, enabling the agent to decompose complex tasks into executable subtasks.
2. The introduction of the Agent-Computer Interface (ACI) is a notable strength of Agent S. This interface serves as a critical abstraction layer that facilitates precise perception and action in GUI environments. By defining a bounded action space with language-based primitives and incorporating a dual-input strategy, ACI enhances the agent's ability to ground actions and receive immediate environmental feedback. This innovation allows Agent S to operate more effectively and efficiently, setting a new standard for MLLM-based GUI agents.

**Weaknesses:**

1. The paper does not address the scalability and efficiency of the framework when handling a large volume of tasks or more complex workflows. There is a need to evaluate how the agent performs under increased load and whether the hierarchical planning and memory update mechanisms can scale without compromising the speed and accuracy of task completion.
2. The framework's performance could potentially falter in scenarios where reliable web knowledge is scarce or when there are frequent, rapid changes in application interfaces that outpace the web's ability to update corresponding information.
3. The paper acknowledges a high rate of execution errors, indicating that Agent S may struggle with decision-making and behavior adjustment during task execution.

**Questions:**

1. Could you elaborate on how Agent S differentiates between when to retrieve external web knowledge versus when to leverage internal memories? How is this decision balanced?
2. How does Agent S handle the cold start problem, especially when it encounters a task that neither memories has prior experience with? Could you explain the strategies for quickly adapting to new tasks?
3. Your error analysis indicates a high rate of execution errors. Could you present a more detailed breakdown of the types of execution errors encountered and discuss potential improvements to the Action Generator to reduce these errors, especially in complex, long-horizon tasks?

---

> ### Author Response · Authors · 2024-11-22
> **Thank you for the review!**
>
> We thank reviewer **meL5** for their positive feedback and insightful comments. We want to take the opportunity to answer their questions and respond to their comments about our work.
>
> `W1: Agent performance under increased load`
>
> Agent S approaches each new task independently, and the volume of tasks does not impact the overall performance. Our agent stores experience memories externally, separate from the language model's context. These memories are retrieved as required. The only source of variation brought about by the increased volume of tasks can be retrieval costs as the KB expands. We precompute all embeddings for saved experiences in advance, so the only overhead during retrieval is calculating the cosine similarity with the current query. Our measurements of retrieval times across different sizes of the experience memory indicate that they remain constant. While our current knowledge base is relatively small, future work could utilize approximate nearest-neighbor algorithms to achieve better retrieval times with larger knowledge bases.
>
> Memory Size (# Exploration Tasks) | 100     | 200     | 300     | 400     |
> |-----------------------------------|---------|---------|---------|---------|
> | Retrieval Time (seconds)          | 0.0417  | 0.0429  | 0.0432  | 0.0446  |
>
>
> | Memory Size (# Exploration Tasks) | Success Rate (%) |
> |-----------------------------------|-------------|
> | 0                                 | 15.38       |
> | 214                               | 23.08       |
> | 427                               | 26.15       |
>
> While complex workflows can affect the performance of our Agent, this is a very general factor, and any Agentic framework attempting complex tasks will suffer from similar issues due to the larger time and steps required for completion.
>
> `W2: Adapting to scarcity of reliable web knowledge and rapid interface changes.`
>
> Scenarios with missing web knowledge and rapidly changing interfaces are some of the key challenges for GUI agents. In situations where web knowledge is missing—such as with new applications or niche interfaces that lack comprehensive online documentation—we employ self-supervised exploration to supplement our agent's understanding. This method does not rely on ground-truth labels; instead, the agent engages in dummy tasks to learn about various workflows within these settings. Through this exploration, the agent can build knowledge about the application interfaces independently, ensuring robust performance even when up-to-date web information is unavailable. The web search and experience retrieval in Agent S distinguishes it from baseline methods, which are bounded by the pre-trained knowledge of MLLMs.
>
> `Q1: Balancing between web search and Experience Retrieval`
>
> Agent S differentiates between retrieving external web knowledge and leveraging internal memories by first formulating a focused query for each task using a Multimodal Large Language Model (MLLM). This query is used to retrieve relevant information from both the web and the agent's internal knowledge base, which contains narrative memories of prior experiences. A Knowledge Fusion Module (MLLM) takes both of these along with the Task instruction as context to generate relevant integrated knowledge. By dynamically balancing external and internal knowledge using an MLLM, Agent S generates an end-to-end plan that effectively utilizes the most appropriate information without over-reliance on either source.
>
> `Q2: Cold-start problem and addressing scenarios where experience is not available`
>
> Agent S addresses the cold start problem by performing self-supervised exploration to populate its Knowledge Base (KB) with information from dummy tasks and tasks derived from existing OSWorld tasks, relying on web knowledge and pre-trained LLM knowledge. Moreover, when encountering a new task with no prior experience, it benefits from multi-level retrieval, as new tasks often include familiar subtasks like opening file menus or closing pop-ups. By performing independent subtask-level retrieval, Agent S can leverage saved experiences from unrelated tasks.
>
> In the most challenging scenarios where neither the task nor its subtasks are related to anything in the KB, the agent relies on the pre-trained knowledge and common-sense reasoning abilities of Multimodal Large Language Models (MLLMs), as well as real-time information from web searches. Additionally, Agent S employs a continual learning mechanism that updates its Knowledge Base upon encountering novel tasks, enabling it to adapt more effectively to new tasks and applications in the future.

---

> > ### Author Response · Authors · 2024-11-22
> > **Detailed breakdown of execution errors and future work**
> >
> > `W3 and Q3: Clarification about the high rate of execution errors and breakdown of the types of execution errors`
> >
> > The error analysis is reflective of only the failed trajectories in our evaluations. Nevertheless, it is true that current MLLM agents are in the early stages and face severe issues with regard to grounding, maintaining long-term states and variables, and improving based on feedback in GUI applications. We did another round of analysis of all the failed cases and further categorized the reasons for failures.  The percentages represent the occurrence of each failure type among all analyzed failed cases from the test_small_new subset. The primary failure reasons identified are
> >
> > | Error Type            | Proportion of Cases |
> > |------------------------|---------------------|
> > | Grounding             | 47.83%             |
> > | Domain Knowledge      | 32.61%             |
> > | Perception            | 21.74%             |
> > | Limited Steps         | 6.52%              |
> > | Feasibility           | 6.52%              |
> > | Problem Understanding | 6.52%              |
> >
> > 1. Grounding Issues: Misalignment between the agent's actions and the intended UI elements, such as misclicks or typing in incorrect fields, or limited granularity of grounding falls under this category. Training more accurate visual grounding models and refining foundation models can improve the agent's ability to interact correctly with UI elements. Engineering more application specific-ACIs can also help mitigate these grounding errors.
> > 2. Domain Knowledge Gaps: The agent lacks highly specific knowledge about certain UI elements or workflows. Enhancing domain knowledge can be achieved by scaling up the system to gather more diverse experiences and by improving the underlying foundation models. This would enable the agent to better understand specialized interfaces and workflows.
> > 3. Perception Errors: These occur when the agent misses details in the observation, leading to incorrect decisions, repeated actions, or misinterpreted feedback. Improving perception involves developing better state representations of observations and employing more advanced foundation models with enhanced visual understanding capabilities.
> > 4. Other categories include the inability of the agent to understand the problem definition, analyze the feasibility or the agent being too slow to finish the task in time. These issues can potentially be rectified through more exploration, training better foundation models and using extra reasoning and verification steps to understand the given task.

---

> > > ### Author Response · Authors · 2024-11-25
> > > **Looking forward to further discussion**
> > >
> > > We thank you for your positive feedback and insightful comments. We appreciate your regard for our work as a groundbreaking framework that stands out for its task automation through experience-augmented hierarchical planning, and the acknowledgment of our ACI as an innovation allows Agent S to operate more effectively and efficiently.
> > >
> > > We are sincerely grateful for the effort you have invested in helping us improve our work. Your feedback on our work has been invaluable, and we have carefully addressed your concerns in our response. With the discussion period deadline approaching, we would greatly appreciate any further comments you might have!

---

> > > > ### Author Response · Authors · 2024-11-27
> > > >
> > > > Dear Reviewer meL5,
> > > >
> > > > As today is the final day for uploading a revision, we wanted to kindly follow up regarding our responses to your comments on our submission.
> > > >
> > > > If there are any additional points or suggestions you’d like us to address, we would be happy to incorporate them before the deadline. We would also greatly appreciate it if you could let us know whether your concerns or questions have been fully addressed.
> > > >
> > > > Thank you once again for your thoughtful feedback and support!
> > > >
> > > > Best regards,
> > > > Authors

---

### Author Response · Authors · 2024-11-22
**Summary of responses and revisions**

We are grateful to all the reviewers for their insightful comments and suggestions for our work. We are pleased that the reviewers found our framework effective and achieving strong performance on benchmarks (**YYG1**, **KnHm**, **7E6Z**); that our experimental evaluation is systematic and comprehensive (**YYG1**, **KnHm**); that our Agent-Computer Interface (ACI) design for GUI control is innovative (**meL5**, **KnHm**); that our novel memory mechanism is effective (**meL5**, **KnHm**); and that the paper is well-presented and easy to follow (**YYG1**, **7E6Z**).

We have made the following revisions based on Reviewers’ feedback:

1. Explanations for Problem definition and hierarchical planning have been added to Section 3, and additional context about UI element grounding is added to Subsection 3.3 to facilitate understanding for new readers
2. Table 1. has been updated to represent that the baselines used in the paper are based on the MMAgent, and other agent baselines (CRADLE, OpenInterpreter, and FRIDAY) have been added.
3. Figure 4 has been updated to focus more on memory construction and learning
4. Figure 8 has been added to the Appendix to illustrate the Agent Computer Interface

We have also added additional results in comments to the Reviewers to answer their questions and address their concerns.

---

### Meta-Review · Area_Chair_VAV9 · 2024-12-16

**Metareview:**

This paper introduces Agent S, a novel framework for controlling GUI-based operating systems, leveraging three core strategies: experience-augmented hierarchical planning, continual memory update, and an Agent-Computer Interface.

The proposed framework is comprehensive and involves the key components of for LLM agents. The work performs comprehensive evaluations across multiple benchmarks and demonstrates the framework’s effectiveness.

Although the proposed framework achieved state-of-the-art performance at the time of submission, the number of baselines is somewhat limited, potentially due to the relative novelty of the benchmarks.

**Additional Comments On Reviewer Discussion:**

The authors addressed the key concerns on the limited scope of baselines.

---

### Decision · Program_Chairs · 2025-01-22

Accept (Poster)